# Masking the Unknown: Leveraging Masked Samples for Enhanced Data Augmentation

Xun Yao[1]      Zijian Huang[1]      Xinrong Hu[1]      Jie (Jack) Yang[2]      Yi Guo[3]

[1] School of Computer Science and Artificial Intelligence, Wuhan Textile University, Hubei Province, China
[2] School of Computing and Information Technology, University of Wollongong, New South Wales, Australia
[3] School of Computer, Data and Mathematical Sciences, Western Sydney University, New South Wales, Australia

## Abstract

Data Augmentation (DA) has become a widely adopted strategy for addressing data scarcity in numerous NLP tasks, especially in scenarios with limited resources or imbalanced classes. However, many existing augmentation techniques rely on randomness or additional resources, presenting challenges in both performance and practical implementation. Furthermore, there is a lack of exploration into what constitutes effective augmentation. In this paper, we systematically evaluate existing DA methods across a comprehensive range of text-classification benchmarks. The empirical analysis highlights that the most significant change resulting from augmentation is observed in the data variance. This observation inspires the proposed approach, termed **M**ask-for-**D**ata **A**ugmentation (**M4DA**), which strategically masks tokens from original samples for augmentation. Specifically, **M4DA** consists of a Variance-Oriented Masker Module (VMM), which ensures an increase in data variances, and a Complexity-Enhanced Selection Module (CSM), designed to select the augmented sample with the highest semantic complexity. The effectiveness of the proposed method is empirically validated across various text-classification benchmarks, including scenarios with limited or full resources and imbalanced classes. Experimental results demonstrate considerable improvements over state-of-the-arts.

## 1 INTRODUCTION

Data Augmentation methods are extensively utilized in Natural Language Processing (NLP), incorporating a range of strategies aimed at enhancing training data. The integration of augmented training samples has proven crucial in achieving remarkable success across numerous downstream tasks, including sentiment analysis [Yoon et al., 2021], question answering [Chen et al., 2022], and review classification [Ren et al., 2021], among others.

Consequently, extensive and diverse research efforts have been devoted to proposing varied strategies for generating diverse and meaningful samples. One approach in this field involves the creation of additional samples at the *character* or *word* level through operations such as swapping, inserting, deleting, replacing original characters/words with alternatives, or perturbing word embeddings [Wei and Zou, 2019, Karimi et al., 2021, Yi et al., 2021, Wei et al., 2021, Ren et al., 2021, Zheng et al., 2023]. Another line of the work focuses on the *sentence* level, employing techniques like translation, mixing, or perturbation of existing sentences to generate variations [Yoo et al., 2021, Yoon et al., 2021, Zhang et al., 2022, Chen et al., 2022, 2023]. Section 2 provides a summary about existing DA approaches.

Despite their prevalence in NLP, existing DA methods suffer from two serious weaknesses. First, a critical concern arises from the inherent randomness observed in certain existing DA methods, wherein candidates (whether characters, words or sentences) are selected and modified in a random manner [Wei and Zou, 2019, Karimi et al., 2021, Wei et al., 2021, Ren et al., 2021, Yoon et al., 2021, Chen et al., 2022]. This randomness introduces a significant risk of incorporating less meaningful or even conflicting augmented data into the training process, thereby exhibiting undesired variations and inconsistencies. Second, computational expenses represent another challenge. Techniques, designed to enhance datasets through sophisticated transformations or synthesis of instances, frequently result in extended processing times and increased infrastructure demands, particularly evident in resource-intensive models like Large Language Models [Yi et al., 2021, Xie et al., 2020, Yoo et al., 2021, Zhang et al., 2022]. This not only limits the scalability of these methods but also poses practical challenges in real-time or resource-constrained environments. Addressing these weaknesses requires approaches that have a comprehensive understanding

towards the data augmentation, which prompts the following critical research question: *What is the primary factor undermining the positive impact of Data Augmentation?* While it is relatively straightforward to attribute the benefits of DA to an expanded data scale, the specific contributing factor that influences the model performance remains unknown.

To tackle it, we conduct a systematic evaluation of existing DA methods across a comprehensive range of text-classification benchmarks. The augmentation performance is assessed using five metrics, with particular emphasis on the metric of **data variance**, which exhibits the most significant change before and after augmentation. This observation motivates the exploration of a novel DA method from the variance change introduced by augmented data. Specifically, this paper introduces **M**ask-for-**D**ata **A**ugmentation, namely **M4DA**, that strategically masks tokens from original samples for augmentation.

It is worth noting that the `[Mask]` token is typically used to hold out a portion of input tokens for predicting missing tokens. Although a limit number of studies leverage `[Mask]` to generate augmented samples [Maas et al., 2011, Gao et al., 2022, Hu et al., 2023, Yao et al., 2023b, Liu et al., 2023], their primary aim still remains filling in masked tokens. In contrast, **M4DA** directly incorporates samples with masked tokens during the model fine-tuning. Yet, the introduction of `[Mask]` tokens brings forth two additional challenges: determining *what content to mask* and *how many tokens to mask*. To address the question of *what content to mask*, we implement random masking, which involves masking the original samples multiple times to generate a diverse candidate pool. Additionally, we leverage *semantic complexity* to identify challenging samples for masking. Regarding the masking budget (*i.e.*, *how many tokens to mask*), we ensure that the masking process is guided to promote an increase in data variance, thereby enhancing the effectiveness of subsequent model training. This approach facilitates a more comprehensive exploration of the data space, leading to improved model robustness and performance. In summary, our contributions are three-fold:

- We perform a comprehensive empirical study of existing Data Augmentation methods utilizing benchmark datasets and evaluating with five metrics, where the empirical analysis reveals that data variance contributes significantly.
- We introduce the `[Mask]`-for-Data-Augmentation approach, referred to as **M4DA**. In this approach, token masking is directed to ensure an increase of data variances and the incorporation of maximum semantic complexity into the augmented data.
- We conduct extensive experiments across diverse benchmarks to demonstrate the superior performance of **M4DA**. Additionally, a series of ablation studies is presented to examine the stability and robustness of our approach.

## 2 RELATED WORK

Data Augmentation (DA) is of great significance as a widely employed technique, which aims at expanding datasets through the generation of modified replicas from existing data or the creation of new instances based on prior knowledge. Embraced extensively in various Natural Language Processing (NLP) tasks, DA has consistently delivered promising outcomes, exemplified by notable successes in various tasks [Yoon et al., 2021, Chen et al., 2022, Ren et al., 2021, Yao et al., 2023a].

One representative direction of DA involves the manipulation of *words* (or *tokens*) within existing textual data. For instance, **EDA** [Wei and Zou, 2019] introduces a collection of augmentation techniques, including synonym replacement and random insertion, swapping, or deletion of words. **AEDA** [Karimi et al., 2021] generates new instances via randomly inserting noisy punctuation, such as colons, periods, and exclamation marks, into the original sentences. In **BERT-aug** [Yi et al., 2021], input tokens are randomly masked out and subsequently replaced using a pre-trained BERT model, resulting in the creation of a new sample. Furthermore, **MTV**[Wei et al., 2021], undertakes token replacement using either synonyms or arbitrary tokens. **TAA** [Ren et al., 2021] also incorporates token replacement (deletion), but leveraging a Bayesian-based optimization algorithm to identify suitable candidate tokens. **AWD** [Chen et al., 2023] uses weighted mixing of word embeddings, prioritizing important words while also enhancing weights of less crucial ones.

An alternative direction for DA focuses at the *sentence* level. **Back-trains** [Xie et al., 2020] employs a two-way model for multilingual translations to generate augmented data. **GPT3Mix** [Yoo et al., 2021] leverages Large Language Models (LLMs), such as GPT-3, to craft high-quality composite samples by amalgamating existing contextual instances. Similarly, **SSMix** [Yoon et al., 2021] alters existing sentences by first identifying important spans based on saliency information and subsequently rewriting them to generate new sentences. Meanwhile, **TreeMix** [Zhang et al., 2022] utilizes constituency parsing trees to segment sentences into components and then recombines them to create new augmentations. In addition, **DoubleMix** [Chen et al., 2022] combines original data with synthetically perturbed data in the hidden space to generate new sentences, while **SEMix** [Zheng et al., 2023] integrates the self-evolution learning with the mixup-based data augmentation.

The summarized overview of discussed DA approaches is presented in Table 1. Despite the prevalence of DA, two main issues persist. First, certain approaches employ random selection strategies to determine candidate words/tokens, potentially introducing noisy augmented data [Wei and Zou, 2019, Karimi et al., 2021, Wei et al., 2021, Ren et al., 2021, Yoon et al., 2021, Chen et al., 2022]. Second, some methods

Table 1: A summary of existing Data Augmentation on the *word* (or *token*) and *sentence* level.

| Index | Method | Operation | | | | | | | Level | |
|---|---|---|---|---|---|---|---|---|---|---|
| | | Swap | Insert | Delete | Replace | Noise | Mixup | Transfer | Word | Sentence |
| I | EDA [Wei and Zou, 2019] | ✓ | ✓ | ✓ | ✓ | | | | ✓ | |
| II | AEDA [Karimi et al., 2021] | | | | | ✓ | | | ✓ | |
| III | BERT-aug [Yi et al., 2021] | | | | ✓ | | | | ✓ | |
| IV | MTV [Wei et al., 2021] | | ✓ | | | | | | ✓ | |
| V | TAA [Ren et al., 2021] | ✓ | ✓ | ✓ | ✓ | | | | ✓ | |
| VI | Back-trans [Xie et al., 2020] | | | | | | | ✓ | | ✓ |
| VII | GPT3Mix [Yoo et al., 2021] | | | | ✓ | | | | | ✓ |
| VIII | SSMix [Yoon et al., 2021] | | | | ✓ | | | | | ✓ |
| IX | TreeMix [Zhang et al., 2022] | ✓ | ✓ | | | | ✓ | | | ✓ |
| X | DoubleMix [Chen et al., 2022] | | | | | ✓ | ✓ | | | ✓ |

necessitate model inference or rely on external resources, such as LLMs, resulting in computational expenses [Yi et al., 2021, Xie et al., 2020, Yoo et al., 2021, Zhang et al., 2022]. In this work, we fill in this gap by presenting a comprehensive empirical evaluation (present in Section 3) and developing a simple-yet-effective DA approach motivated by our observations (present in Section 4).

# 3 REVISITING DATA AUGMENTATION

The research interest in leveraging Data Augmentation (DA) for downstream applications is particularly significant in low-resource and/or class-imbalanced settings. Yet, a noticeable gap exists regarding the contributing factors or information embedded within augmented data. Specifically, the exploration of how certain characteristics influence the efficacy of augmentation remains largely unexplored. In this section, we attempt to address this gap by examining the effects of DA from a statistical perspective.

## 3.1 EFFECT OF AUGMENTATIONS

Let $\mathcal{X} \supset X = [X^{\mathrm{ORG}}, X^{\mathrm{AUG}}]$, where $X^{\mathrm{ORG/AUG}}$ represents the set of original/augmented samples, and $\mathcal{X}$ is the input space. Subsequently, we further obtain a **naive** model (encoder) $\mathcal{M} : \mathcal{X} \to \mathcal{Z}$, fine-tuned exclusively with $X^{\mathrm{ORG}}$, where $\mathcal{Z}$ is the output (feature) space. The **augmented** model, written as $\mathcal{M}'$, is trained using both $X^{\mathrm{ORG}}$ and $X^{\mathrm{AUG}}$. Further, let $Z, Z' \subset \mathcal{Z}$ be the hidden representations of $X$ obtained from $\mathcal{M}$ and $\mathcal{M}'$, respectively (*e.g.*, the last-layer *[CLS]* embedding in Transformer-based encoders). Accordingly, $z_i, z_i' \in \mathcal{Z}$ denote the $i$-th representations from $Z$ and $Z'$, *e.g.*, $z_i^{(')} = \mathcal{M}^{(')}(x_i)$. This study employs the following widely-recognized metrics to access changes before and after the augmentation, including:

- **Variance:** a measure of how the data spreads out or scatters, *i.e.*, $\mathrm{Var}(Z/Z')$: $\mathrm{Var}(Z) = \frac{1}{L} \sum_l^L var(Z^l)$, where $var$ is the variance operator and $Z^l$ is the $l$th dimension with slight abuse of notations.
- **Covariance:** the data interdependence or correlation level,

*i.e.*, $\mathrm{Cov}(Z/Z')$: $\mathrm{Cov}(Z) = \frac{2}{L(L-1)} \sum_{l>m} cov(Z^l, Z^m)$ where $cov$ is the covariance operator.

- **Pair Invariance:** the robustness between $Z$ and $Z'$, irrespective of the applied augmentation, estimated by the mean-squared Euclidean distance and cosine similarity:

$$\begin{cases} \mathrm{PI}_{\mathrm{MSE}}(Z, Z') = \frac{1}{n} \sum_{i=1}^n \|z_i - z_i'\|_2^2, \\ \mathrm{PI}_{\mathrm{COS}}(Z, Z') = \frac{1}{n} \sum_{i=1}^n \frac{z_i \cdot z_i'}{\|z_i\|\|z_i'\|}. \end{cases} \quad (1)$$

- **Effective invariance [Gao et al., 2022]:** the measure of consistency and confidence of predictions:

$$EI(Z, Z') = \begin{cases} \sqrt{p_Z \cdot p_{Z'}} & \text{if correct} \\ 0 & \text{otherwise.} \end{cases} \quad (2)$$

where $p_{Z/Z'}$ is prediction confidence score using $Z$ or $Z'$.

The effect of various augmentations (discussed in Section 2) is investigated via following the experimental setup outlined in [Karimi et al., 2021, Yi et al., 2021, Wei et al., 2021, Ren et al., 2021], with additional details provided in Appendix A. The resulting metrics for the SST-2 dataset, over 10 runs, are presented in Fig. 1, and again, experimental results on other datasets are provided in Appendix A. Additionally, Bayesian hypothesis testing [Rouder et al., 2009] is incorporated to ensure the reliability of our findings. This is accomplished by computing the Bayes factor $\mathrm{BF}_{10}$ to gauge the degree of support for the alternative hypothesis over the null hypothesis. In this context, the alternative hypothesis $\mathcal{H}_1$ states that *there is sensitivity to augmentation in terms of Covariance and (Pair/Effective) Invariance*, while the null hypothesis $\mathcal{H}_0$ suggests *there is no discernible difference*.

The following observation is made. 1) As expected, the model trained with augmented data consistently demonstrates superior accuracy compared to its naive counterpart. 2) A clear trend is the general *rise in variance*, that is proportional to the performance gain. 3) In contrast with the variance metric, there is typically no substantial change in the covariance and (pair/effective) invariance. For example, while two methods (*e.g.*, V and VI) may exhibit similar augmentation gains, they might yield entirely different covariance values. A similar occurrence is noted with

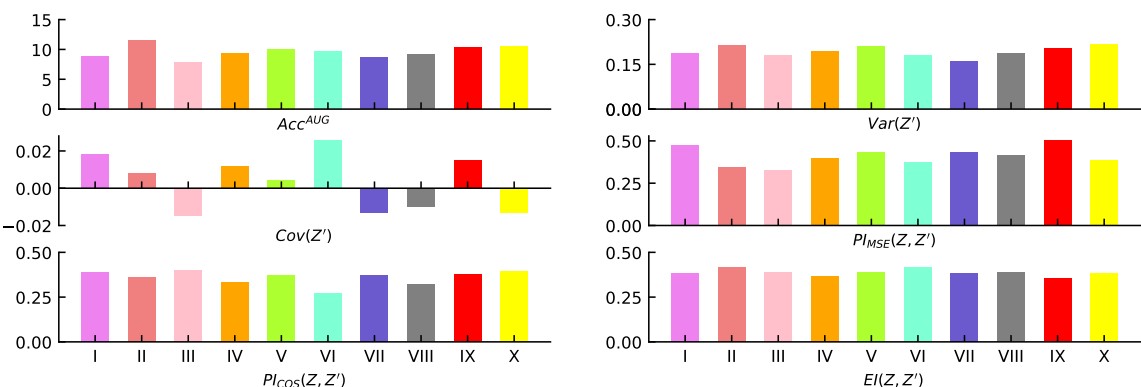

Figure 1: The resulting metrics over 10 runs on the SST-2 dataset. The **naive** model $\mathcal{M}$ achieves the classification accuracy of 68.3±3.7, $\text{Var}(Z)$ is 0.138±0.121, and $\text{Cov}(Z)$ is 0.372±0.051. The three plots labelled by $Acc^{\text{AUG}}$, $\text{Var}(Z')$, and $\text{Cov}(Z')$ respectively represent the differences between the **augmented** model $\mathcal{M}'$ and the **naive** model $\mathcal{M}$.

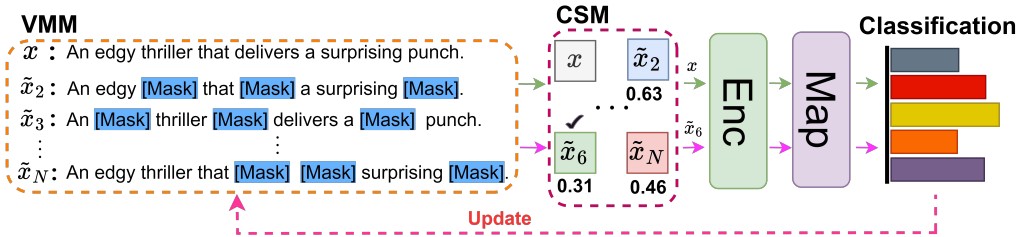

Figure 2: Illustration of the proposed **M4DA**, maximizing data variance and semantic complexity of augmented samples.

(pair/effective) invariance. Consequently, utilizing covariance and invariance as an augmentation indicator is not feasible. Bayesian hypothesis testing also supports these observations, consistently rejecting the $\mathcal{H}_1$ hypothesis with a robust BF10 consistently exceeding 100. Motivated by this observation, a novel augmentation method with a specific focus on data variance is introduced in the next section.

# 4 PROPOSED METHOD

In this section, we introduce our proposed **M**ask-for-**D**ata **A**ugmentation (**M4DA**) method. The overall pipeline is illustrated in Fig. 2. **M4DA** consists of two key modules, *i.e.*, Variance-oriented Masker Module (VMM, addressing the issue of **how many tokens to mask**) and Complexity-enhanced Selection Module (CSM, addressing the issue of **what content to mask**). Specifically, original inputs are masked to generate additional training samples, with a simultaneous emphasis on maximizing data variance (via VMM) and semantic complexity (vis CSM) of augmented data to ensure the effectiveness. In the following, we will introduce the design of **M4DA** in detail.

## 4.1 VARIANCE-ORIENTED MASKER MODULE(VMM)

Given the tokenized original sequence $\boldsymbol{x}$, for the classification task, we aim to optimize an encoder $\mathcal{M}_\phi$ with parameters in $\phi$ to extract the latent representation $\boldsymbol{z} = \mathcal{M}(\boldsymbol{x})$ and a Multilayer Perceptron (MLP) layer (with all its parameters collected in $\theta$) $f_\theta : \mathcal{Z} \to \mathcal{Y}$ to *map* to its target label $y \in \mathcal{Y}$ such that, $y = f(\boldsymbol{z})$. We drop the parameters when it is clear. The encoder $\mathcal{M}_\phi$ is typically fine-tuned with a penalty function, such as the c̲ross-e̲ntropy-based loss.

To augment $\boldsymbol{x}$, conventional approaches primarily rely on data perturbation, including methods like synonym replacement and word insertion/swapping/deletion. Yet, the perturbation task is nontrivial, particularly in the textual context, where directly modifying input texts can result in semantically meaningless or even conflicting augmentations [Pellicer et al., 2023].

To address this issue, we introduce the variance-oriented masker module, strategically substituting existing tokens in $\boldsymbol{x}$ with [Mask]. In this module, we introduce $\eta$, a masking budget (*i.e.*, the fraction of masked tokens). For an original instance $\boldsymbol{x} \in X^{\text{ORG}}$, we apply $T = \lceil |\boldsymbol{x}| * \eta \rceil$ [Mask] tokens to randomly replace existing ones within $\boldsymbol{x}$ and generate a masked sequence $\tilde{\boldsymbol{x}}$. We repeat this random-

ized masking $N$ times, resulting in an augmentation set $\tilde{X} = \{\tilde{\boldsymbol{x}}_1, \tilde{\boldsymbol{x}}_2, \cdots, \tilde{\boldsymbol{x}}_N\}$, whose items will be screened by the selection module.

## 4.2 COMPLEXITY-ENHANCED SELECTION MODULE(CSM)

The variance-oriented masker module introduces randomness by masking existing tokens in input sequences, resulting in augmented samples with increasing variances. However, the lack of control in this random masking can lead to arbitrary outcomes. To address this limitation, we introduce the complexity-enhanced selection module, with the objective of choosing the augmented sample with the highest semantic complexity. This enhancement is intended to potentially benefit the subsequent effectiveness of encoder training.

For this purpose, we utilize the Kantorovich-Rubinstein Distance [Villani, 2016] to select *one* augmented sample from $\tilde{X}$ with the maximum distance to the cohort for the final augmentation. Let $(\mathcal{X}, \gamma)$ be a Polish metric space with metric $\gamma(x, y)$ where $\boldsymbol{x}_i$ or $\tilde{\boldsymbol{x}}_i$ are from. For any two probability measures $\mu, v$ on $\mathcal{X}$, the Kantorovich-Rubinstein distance between $\mu$ and $v$ is defined by

$$
\begin{aligned}
W(\mu, v) = \inf_{p \in \Pi(\mu, v)} \int_{\mathcal{X}} \gamma(x, y) dp(x, y), \\
\text{s.t.} \int_{\mathcal{X}} p(x, y) dy = \mu, \int_{\mathcal{X}} p(x, y) dx = v.
\end{aligned}
\tag{3}
$$

where $\Pi(\mu, v)$ is the set of all couplings of $\mu$ and $\nu$. We use token frequency as $\mu$ and $\nu$. In particular, $\mu$ is for $\tilde{X}$ and $\nu$ is for the candidate from $\tilde{X}$. $\gamma$ is the pairwise token distance of their vector representations, say from BERT tokenizer. Basically, we select the one that has the minimum semantic coverage among $\tilde{X}$, equivalent to the highest semantic complexity. The computation of Eq. (3) is carried out by Sinkhorn algorithm Cuturi [2013] efficiently.

## 4.3 OPTIMIZATION

The proposed CSM module collaborates with VMM to generate a most effective augmented instance from a given one. Drawing inspiration from the insights in Section 3, the process is guided through iterative optimization, with the objective of achieving progressive variance increase.

In simpler terms, our optimization process is an iterative scheme that alternates between enhancing performance and increasing variance, with both CSM and VMM embedded within it. To this end, we introduce a probability generator $p_\omega$ with parameters collected in $\omega$ such that

$$
\eta = p_\omega(\mathcal{M}_\phi(X)).
\tag{4}
$$

As a result, the masking probability is adaptive and subject to optimization. Since we compute the variance in $\mathcal{Z}$, it is directly linked to $\mathcal{M}$, specifically through its parameter set $\phi$. Whenever we update $\phi$, the variance will consequently change. We start with an encoder $\mathcal{M}_{\phi^0}$, such as a pre-trained BERT model with parameters $\phi^0$. Suppose at step $k$, let $X$ and $Y$ represent the original samples and their labels, respectively, and $\tilde{X}^k$ denote the corresponding augmented samples generated through VMM and CSM using $\mathcal{M}_{\phi^k}$ and $\eta^k$. Note that $\eta^k = p_{\omega^k}(\mathcal{M}_{\phi^k}(X))$ by Eq. (4). We define $D^k = \{X \cup \tilde{X}^k, Y \cup Y^k\}$ as the combined data set at step $k$, noting that $Y^k = Y$. Subsequently, we perform the following to update the encoder and the MLP layer as:

$$
(\theta^{k+\frac{1}{2}}, \phi^{k+\frac{1}{2}}) = \underset{\phi, \theta | \phi^k, \theta^k}{\arg\min} \mathcal{L}_{ce}(f_\theta \circ \mathcal{M}_\phi, D^k), \tag{5}
$$

then

$$
\begin{aligned}
(\omega^{k+1}, \theta^{k+1}, \phi^{k+1}) = \underset{\omega, \theta, \phi | \omega^k, \phi^{k+\frac{1}{2}}, \theta^{k+\frac{1}{2}}}{\arg\min} \frac{\text{Var}(\mathcal{M}_{\phi^{k+\frac{1}{2}}}(\tilde{X}^k))}{\text{Var}(\mathcal{M}_\phi(\tilde{X}^k))} \\
+ \mathcal{L}_{ce}(f_\theta \circ \mathcal{M}_\phi, D^k),
\end{aligned}
\tag{6}
$$

where $\circ$ is functioncomposition, $\mathcal{L}_{ce}(f, D)$ represents the cross-entropy loss for model $f$ on data $D$, and $\arg\min_{\xi | \xi^k} \mathcal{L}$ means minimizing $\mathcal{L}$ *w.r.t* $\xi$ starting with $\xi^k$. The minimization process involves a single iteration using gradient descent, as chosen in our experiments. Eq. (5) aims to boost the classification performance, while Eq (6) aims to increase the variance before and after the augmentation, with the updated encoder and the MLP layer. As the optimization progresses, an increase in variance is expected, as also confirmed by our experiments. Furthermore, the utility of augmented data is maximized throughout the process, with *intermediate* samples utilized for training the model. The optimization process terminates upon reaching a predetermined maximum number of steps (epochs).

## 5 EXPERIMENTS

### 5.1 DATASETS

For the text classification task, we employ the following benchmarks: (1) IMDB [Maas et al., 2011], SST-2, and SST-5 [Socher et al., 2013] for sentiment classification, (2) TREC [Li and Roth, 2002] for question-type classification, (3) YELP-2 and YELP-5 [Zhang et al., 2015] for review classification. The dataset statistics are provided in Table 2.

**Low-resource setting.** We downsample each dataset to create smaller training and validation sets while preserving the original distribution using *Stratified Shuffle Split* [Shahrokh Esfahani and Dougherty, 2013]. The resulting datasets (IMDB, SST-5, SST-2, TREC, YELP-2, and YELP-5) contain different numbers of labeled training

Table 2: Statistics on employed text classification benchmarks, where #Classes, #Train, #Test, #Min, #Max, #Median, and #Length represent, respectively, the number of classes, instances in the training and test sets, minimum length, maximum length, median length, and the average length of input text.

| Dataset | #Classes | #Train | #Test | #Min | #Max | #Median | #Length |
|---------|----------|--------|-------|------|------|---------|---------|
| IMDB | 2 | 25,000 | 25,000 | 2 | 2470 | 173 | 326 |
| SST-2 | 2 | 7,791 | 1,821 | 2 | 56 | 18 | 18 |
| YELP-2 | 2 | 560,000 | 38,000 | 1 | 1052 | 97 | 139 |
| TREC | 6 | 5,452 | 500 | 3 | 37 | 9 | 10 |
| SST-5 | 5 | 9,643 | 2,210 | 2 | 56 | 18 | 19 |
| YELP-5 | 5 | 650,000 | 50,000 | 1 | 1052 | 99 | 141 |

samples (80, 200, 80, 120, 80, and 200, respectively) and validation samples (60, 150, 60, 60, 60, and 150, respectively). To ensure fair comparison, we only consider one augmented sample for every existing one.

**Class-imbalanced setting.** Following the experimental setup outlined in [Ren et al., 2021], for binary classification datasets like IMDB, YELP-2, and SST-2, we reduce positive samples from the training set. That is, the negative class within the training set comprises 1000 samples, while the positive class contains only 20/50 training samples, denoted respectively as $\gamma_{\text{imb}} = 2\%$ and $\gamma_{\text{imb}} = 5\%$ (the ratio between positive and negative samples). Additionally, we utilize the Over-Sampling (OS) baseline in this scenario, duplicating positive class training samples 50 times for an imbalance ratio of 2% and 20 times for a ratio of 5%.

## 5.2 BASELINES AND EVALUATION

Our method is compared with the following baselines, including **EDA** [Wei and Zou, 2019], **Back-trans** [Xie et al., 2020], **BERT-aug** [Yi et al., 2021], **AEDA** [Karimi et al., 2021], **TAA** [Ren et al., 2021], **MTV** [Wei et al., 2021], **Double Mix** [Chen et al., 2022], **AWD** [Chen et al., 2023] and **SEMix** [Zheng et al., 2023]. Those methods have been discussed in Section 2. For all baselines, we reproduce their results using open-sourced codes with the default parameter settings from the corresponding papers. For **M4DA**, we employ the *BERT-Base* as the embedding encoders (the impact from the encoder type is provided in the ablation study). We utilize an Adam optimizer with a learning rate of $4 \times 10^{-5}$. The training epoch is 20 for TREC and 10 for the remaining datasets. All models are fine-tuned using a NVIDIA A100 GPU server. In low-resource settings, we assess methods using Accuracy(%). For imbalanced scenarios, evaluation includes both Accuracy(%) and F1-score, where higher values signify better classification performance. At last, we conduct all experiments using three random seeds and perform five runs under each seed, and report the mean Accuracy and its standard deviation.

## 5.3 MAIN RESULTS

Table 3 presents an experimental comparison across six text classification tasks in the low-resource scenario. From the results, we derive the following insights: 1) In low-resource contexts, our method consistently outperforms baselines by a significant margin, showcasing the effectiveness of **M4DA** in producing augmented samples that enhance subsequent classification models. 2) Methods such as EDA and BERT-aug, relying on random augmentation strategies, exhibit larger variations in standard deviation due to their dependency on specific parameters. Conversely, heuristic-based methods like AEDA and TAA overlook sample difficulty, employing fixed strategies for augmentation, resulting in suboptimal performance. In contrast, our method is more adaptive and proficient in generating increasingly-complex augmented samples, aligning with the model's learning progress. 3) Even with the original full datasets, our method consistently surpasses existing baseline augmentations across all datasets. This is exemplified by an absolute-point accuracy increase of 2.2 compared to the best baseline (AWD).

Table 4, on the other hand, presents the classification results under the class-imbalance scenario. The following observations are made: 1) All methods generally perform better in a less imbalanced scenario, as evidenced in SST-2, where there is an average accuracy increase of 9.7% when the imbalance ratio is raised from 2% to 5%. 2) Similar to the low-resource scenario, **M4DA** significantly enhances accuracy by an average of approximately 18.9%, surpassing alternative methods. F1-score results in the imbalanced scenario are available in Appendix B.

Overall, results from Table 3 and 4 highlight the superiority of **M4DA**. Other methods do not exhibit a clear advantage in either the low-resource or class-imbalanced scenarios. For instance, while EDA achieves the second-best results in the low-resource setting, DoubleMix is the second place in the imbalanced scenario. Their inconsistent performance can be attributed to either inherent data randomness or the inflexibility in adapting to data difficulty. In contrast, our method consistently outperforms others in both settings, demonstrating its effectiveness in accurately classifying intents by producing effective augmentations.

## 6 ABLATION STUDY

**On the encoder flexibility.** We begin by assessing the impact of the underlying encoder. Specifically, we utilize the RoBERTa-Base encoder [Zhuang et al., 2021]. Most experimental settings, such as batch size and sequence length, remain consistent with previous evaluations, except for the learning rate, which is set to $3e^{-5}$. Comparison results are presented in Table 5 and Table 6, where our method still demonstrates the highest performance across all datasets.

Table 3: Comparison of classification accuracy with the **low-resource** setting. The best and the second best results are indicated in bold and underline, respectively. The first and second number represents the performance obtained from the low-resource and full dataset, respectively. Statistically significant gains achieved by the proposed method at $p$-values $< 0.01$ are marked with †.

| Method | IMDB | SST-2 | SST-5 | TREC | YELP-2 | YELP-5 |
|---|---|---|---|---|---|---|
| BERT-Base | 64.7±3.4/87.5±0.1 | 67.6±4.2/91.2±0.1 | 36.1±3.9/51.8±0.1 | 69.3±5.3/97.0±0.1 | 73.8±4.2/96.1±0.1 | 36.6±4.6/65.1±0.2 |
| +EDA | 72.6±4.6/87.8±0.1 | 76.8±3.3/91.8±0.1 | 37.9±1.2/51.8±0.2 | 82.4±2.6/97.1±0.3 | 74.6±2.4/95.6±0.2 | 36.9±3.1/65.2±0.1 |
| +Back-trans | 73.9±4.3/87.5±0.1 | 77.7±3.1/91.3±0.1 | 35.2±2.1/51.2±0.2 | 79.0±3.8/96.9±0.4 | 77.4±3.3/95.4±0.2 | 40.3±4.3/65.1±0.1 |
| +BERT-aug | 75.9±3.3/87.3±0.1 | 74.4±6.8/91.2±0.1 | 35.9±1.5/51.5±0.2 | 78.6±2.3/97.0±0.1 | 77.4±2.2/95.1±0.1 | 43.4±3.8/65.3±0.1 |
| +AEDA | 75.3±3.5/87.9±0.2 | 79.5±2.5/91.0±0.2 | 37.1±2.3/52.4±0.1 | 80.2±6.3/97.6±0.1 | 78.4±3.7/95.7±0.1 | 41.2±3.0/65.9±0.2 |
| +TAA | 74.2±2.4/88.3±0.1 | 78.0±3.4/91.9±0.2 | 37.4±2.6/52.4±0.2 | 79.6±4.6/97.1±0.1 | 78.9±2.7/96.1±0.1 | 44.6±1.8/65.6±0.2 |
| +MTV | 76.3±4.2/87.5±0.1 | 77.3±1.3/91.5±0.1 | 36.4±1.4/52.6±0.1 | 77.1±3.6/96.8±0.2 | 78.4±2.4/95.8±0.2 | 40.2±3.0/65.4±0.1 |
| +Double Mix | 75.8±2.6/88.0±0.3 | 78.5±2.1/92.1±0.1 | 37.7±1.6/52.9±0.2 | 82.1±1.3/97.4±0.1 | 79.6±3.0/96.0±0.2 | 43.5±3.6/65.5±0.1 |
| +AWD | 77.4±3.6/88.2±0.1 | 81.2±3.9/92.2±0.3 | 39.6±2.8/52.8±0.1 | 83.7±4.4/97.3±0.2 | 80.5±3.9/96.1±0.1 | 45.3±4.1/66.0±0.2 |
| +SEMix | 76.8±3.1/88.1±0.1 | 81.0±3.7/92.0±0.2 | 38.4±2.2/52.7±0.2 | 83.0±3.7/97.3±0.3 | 81.4±2.4/96.2±0.1 | 46.0±4.0/65.8±0.1 |
| **+M4DA** | **79.6**±2.1†/**88.5**±0.2† | **82.3**±1.9†/**92.4**±0.1† | **42.5**±1.5†/**53.1**±0.2† | **84.8**±1.8†/**97.7**±0.1† | **83.2**±1.5†/**96.3**±0.2† | **48.4**±2.7†/**66.1**±0.2† |

Table 4: Comparison of the classification accuracy with the **class-imbalance** setting. The best and second-best results are highlighted in bold and underline, respectively. Statistically significant gains achieved by the proposed method at $p$-values $< 0.01$ are marked with †.

| Method | SST-2 | | IMDB | | YELP-2 | |
|---|---|---|---|---|---|---|
| | $\gamma_{\mathrm{imb}} = 2\%$ | $\gamma_{\mathrm{imb}} = 5\%$ | $\gamma_{\mathrm{imb}} = 2\%$ | $\gamma_{\mathrm{imb}} = 5\%$ | $\gamma_{\mathrm{imb}} = 2\%$ | $\gamma_{\mathrm{imb}} = 5\%$ |
| BERT-Base | 50.2±1.3 | 55.0±6.3 | 50.1±0.0 | 51.8±4.6 | 52.2±1.2 | 56.2±2.6 |
| +OS | 52.0±1.8 | 58.6±5.6 | 52.3±2.7 | 59.1±6.3 | 54.4±2.6 | 61.3±3.1 |
| +EDA | 52.9±3.9 | 59.2±5.9 | 57.3±5.7 | 64.1±7.2 | 59.3±2.2 | 68.6±3.4 |
| +Back-trans | 54.1±4.2 | 59.5±5.7 | 52.2±2.9 | 57.4±6.9 | 62.6±3.5 | 73.2±4.8 |
| +BERT-aug | 54.6±3.2 | 64.9±3.7 | 60.8±2.2 | 68.5±3.8 | 63.7±1.6 | 79.4±3.3 |
| +AEDA | 53.4±1.8 | 67.8±3.9 | 57.8±1.2 | 70.2±5.4 | 60.6±1.7 | 80.2±4.0 |
| +TAA | 56.5±3.7 | 66.1±4.9 | 56.9±2.8 | 66.7±4.2 | 65.8±2.8 | 77.5±4.5 |
| +MTV | 52.6±2.5 | 60.3±4.8 | 54.4±2.2 | 62.4±3.4 | 61.1±2.5 | 72.6±4.8 |
| +Double Mix | 55.1±2.5 | 68.5±3.9 | 59.3±2.1 | 69.6±4.0 | 63.8±1.9 | 81.9±2.7 |
| +AWD | 55.8±2.9 | 69.2±3.1 | 59.9±2.0 | 71.9±5.3 | 65.0±2.5 | 83.1±2.8 |
| +SEMix | 56.2±2.7 | 68.8±3.3 | 60.4±2.5 | 70.4±4.5 | 64.1±2.7 | 82.5±2.3 |
| **+M4DA** | **58.4**±1.6† | **70.3**±4.4† | **61.2**±2.5† | **73.4**±3.8† | **67.4**±1.3† | **84.9**±3.1† |

Additionally, F1-score results in the imbalanced scenario are provided in Appendix B. Furthermore, we observe inconsistent performance among other methods. For instance, when transitioning the encoder from BERT to RoBERTa, the EDA method shows improved performance in the low-resource scenario but exhibits inferior performance in the imbalanced case. These findings demonstrate the stability and robustness of our method across underlying encoders (both RoBERTa and BERT), consistently outperforming current state-of-the-art models. An ablation study assessing the impact of the model size is also conducted, utilizing RoBERTa-Large as the encoder, and the result is provided in Appendix C. To maintain consistency, subsequent ablation studies are conducted using BERT-base.

**On the model breakdown.** We introduce three variants to analyze **M4DA**: +[Mask], which augments the vanilla model with random token masking; +VMM, which augments samples with progressively increasing variance; and +CSM, which further selects augmented samples with the highest semantic complexity. We repeat all variants for five runs with three random seeds, and the results are shown in Fig. 3. Notably, both components of +VMM and +CSM exhibit improvements over the baseline. In particular, the +VMM achieves

a notable increase with an average of 2.3%. However, the +CSM component contributes the most, with a further 3.5% improvement. This comparison highlights the significant contribution of VMM to ensuring performance gains.

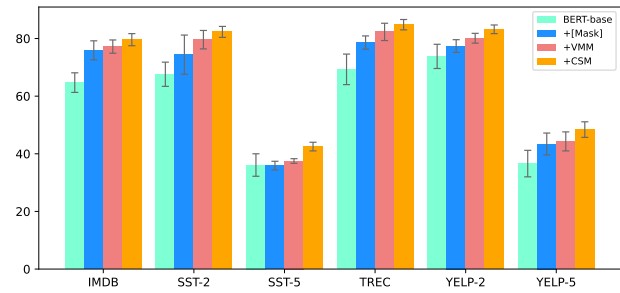

Figure 3: Ablation results on six datasets are presented to evaluate the impact of [Mask] based Data Augmentation, VMM, and optimal transport.

**Variance from augmentation.** We also visualize the variance of augmented samples for each epoch across six datasets in the low-resource case, with results shown in Fig. 4. Taking the SST-5 dataset as an example, during the

Table 5: Comparison of classification accuracy with the **low-resource** setting with the **RoBERTa-Base**. The best and the second best results are indicated in bold and underline, respectively. The first and second number represents the performance obtained from the low-resource and full dataset, respectively. Statistically significant gains achieved by the proposed method at $p$-values $< 0.01$ are marked with †.

| Method | IMDB | SST-2 | SST-5 | TREC | YELP-2 | YELP-5 |
|---|---|---|---|---|---|---|
| RoBERTa-Base | 73.5±0.4/95.0±0.1 | 71.3±1.3/94.6±0.1 | 33.4±2.8/57.1±0.5 | 72.8±5.3/97.1±0.1 | 77.8±1.5/95.7±0.1 | 40.3±3.3/65.4±0.1 |
| +EDA | 84.2±1.5/94.7±0.2 | 82.8±5.8/94.7±0.1 | 40.9±3.9/57.2±0.5 | 82.2±6.9/97.0±0.2 | 88.4±1.9/95.9±0.1 | 46.1±4.2/65.8±0.1 |
| +Back-trans | 83.8±0.3/95.3±0.1 | 83.9±1.8/94.5±0.1 | 40.1±3.2/57.5±0.3 | 81.3±6.2/97.0±0.2 | 87.4±2.5/95.8±0.1 | 47.9±2.6/65.7±0.3 |
| +BERT-aug | 84.2±1.8/95.3±0.1 | 82.9±2.6/94.2±0.1 | 36.9±3.2/56.4±1.4 | 80.9±3.0/97.2±0.4 | 89.1±3.3/95.7±0.2 | 43.7±6.5/65.9±0.1 |
| +AEDA | 82.9±1.1/95.4±0.1 | 83.9±2.3/94.1±0.2 | 38.3±2.1/57.4±0.6 | 82.1±7.0/97.2±0.1 | 87.8±1.7/96.0±0.2 | 46.6±3.9/66.1±0.1 |
| +TAA | 84.0±1.2/95.1±0.1 | 83.2±2.7/94.9±0.1 | 40.3±1.7/57.3±0.4 | 81.9±4.3/97.9±0.1 | 89.3±2.6/96.1±0.2 | 47.5±2.8/66.0±0.2 |
| +MTV | 83.5±1.7/95.2±0.1 | 82.2±2.9/94.4±0.1 | 36.6±6.0/57.7±0.3 | 80.6±4.8/97.8±0.1 | 88.9±1.4/95.6±0.2 | 47.8±2.5/66.2±0.1 |
| +Double Mix | 84.6±0.9/95.0±0.1 | 83.5±1.6/94.8±0.1 | 41.6±2.6/57.5±0.5 | 81.5±3.6/98.0±0.1 | 89.6±1.6/95.9±0.2 | 46.9±2.7/65.9±0.2 |
| +AWD | 85.1±1.2/95.3±0.2 | 85.2±2.4/94.7±0.2 | 42.2±2.3/57.8±0.2 | 83.9±4.7/98.1±0.3 | 90.1±2.6/96.0±0.3 | 47.7±1.9/66.4±0.3 |
| +SEMix | 84.3±1.4/95.2±0.1 | 84.6±2.0/94.7±0.1 | 42.7±3.4/57.4±0.3 | 82.2±3.2/98.0±0.1 | 90.8±2.4/95.8±0.2 | 47.2±2.8/66.2±0.2 |
| **+M4DA** | **86.2**±0.9†/**95.6**±0.1† | **86.3**±2.1†/**95.0**±0.2† | **44.8**±1.8†/**57.9**±0.3† | **84.7**±2.2†/**98.3**±0.2† | **92.7**±1.4†/**96.2**±0.1† | **49.5**±1.8†/**66.6**±0.2† |

Table 6: Comparison of the classification accuracy with the **class-imbalance** setting with the **RoBERTa-Base**. The best and second-best results are highlighted in bold and underline, respectively. Statistically significant gains achieved by the proposed method at $p$-values $< 0.01$ are marked with †.

| Method | SST-2 $\gamma_{\mathrm{imb}} = 2\%$ | SST-2 $\gamma_{\mathrm{imb}} = 5\%$ | IMDB $\gamma_{\mathrm{imb}} = 2\%$ | IMDB $\gamma_{\mathrm{imb}} = 5\%$ | YELP-2 $\gamma_{\mathrm{imb}} = 2\%$ | YELP-2 $\gamma_{\mathrm{imb}} = 5\%$ |
|---|---|---|---|---|---|---|
| RoBERTa-Base | 52.3±1.7 | 57.3±3.8 | 51.3±0.8 | 53.6±1.5 | 52.8±1.0 | 57.6±2.4 |
| +OS | 54.5±2.1 | 63.2±4.3 | 52.5±1.6 | 62.7±2.7 | 56.0±2.3 | 65.8±3.8 |
| +EDA | 54.9±1.6 | 65.3±3.7 | 53.3±1.4 | 64.6±4.5 | 66.3±1.5 | 76.6±5.5 |
| +Back-trans | 55.6±2.6 | 67.3±2.6 | 57.3±2.8 | 67.7±5.8 | 72.6±2.0 | 79.3±3.8 |
| +BERT-aug | 57.2±3.4 | 70.0±4.1 | 56.4±3.3 | 66.1±3.9 | 64.3±2.5 | 78.3±6.5 |
| +AEDA | 63.1±6.0 | 76.6±6.2 | 60.6±2.7 | 72.3±4.9 | 67.6±2.2 | 79.6±4.5 |
| +TAA | 61.5±4.9 | 72.3±4.1 | 62.7±2.6 | 69.7±3.3 | 68.3±3.4 | 81.2±3.8 |
| +MTV | 59.5±2.0 | 68.7±6.1 | 58.8±3.2 | 70.4±4.6 | 67.7±2.3 | 75.6±4.8 |
| +Double Mix | 60.3±2.4 | 74.0±3.6 | 61.1±1.3 | 73.8±2.6 | 70.3±1.3 | 80.2±3.8 |
| +AWD | 62.9±3.3 | 78.2±5.3 | 62.4±1.8 | 76.6±4.3 | 73.7±2.6 | 82.8±4.5 |
| +SEMix | 61.4±2.6 | 77.1±4.3 | 62.0±2.7 | 75.4±3.2 | 74.4±2.9 | 81.9±3.9 |
| **+M4DA** | **65.7**±3.6† | **80.1**±5.0† | **63.5**±2.3† | **78.7**±3.7† | **76.8**±2.3† | **84.6**±5.4† |

first five epochs, the variance values exhibit an increasing trend: $\{0.04, 0.15, 0.28, 0.32, 0.36\}$. This indicates the effectiveness of our strategy in promoting data variance.

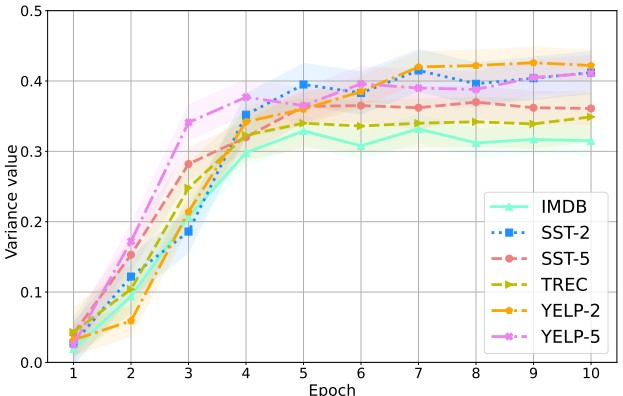

Figure 4: The variance trend of augmented samples at each epoch from six datasets.

**On the masking ratio.** In this experiment, we adjust the

masking rate $\eta$ of augmented samples during training across six datasets in the low-resource scenario, illustrating its variability within specific ranges. The results are summarized in Table 7. For example, the YELP2 dataset exhibits a wide range of masking rates, from a minimum of 38% to a maximum of 68%, indicating significant variation in augmentation. Conversely, the IMDB dataset shows a narrower range (from 20% to 39%). Obviously, the masking rate $\eta$ is determined by the employed data itself, with further discussion provided in Appendix D.

Table 7: The masking rate variation of augmented samples across six datasets.

| | IMDB | SST2 | SST5 | TREC | YELP2 | YELP5 |
|---|---|---|---|---|---|---|
| Min | 20 % | 26 % | 28 % | 28 % | 38 % | 20 % |
| Max | 39 % | 43 % | 42 % | 44 % | 68 % | 59 % |
| Avg | 32 % | 31 % | 37 % | 39 % | 56 % | 40 % |

**On the augmented samples.** M4DA augments one original sample ten times to create ten candidates, which are then selected using the CSM during training. To examine the

impact of the number of augmented samples, we vary the number of augmentations to 5, 10, 15, and 20. The results are presented in Table 8. Clearly, providing more augmented samples allows the model to choose better augmentations. For example, setting the number of augmented samples to 10 yields an average improvement of 1.4% over 5 across all six datasets. However, when using 15 candidates, the increase is insignificant, indicating that moderate value (*i.e.*, 10) typically results in commendable performance.

Table 8: The ablation results on six datasets to evaluate the impact from the numbers of augmented samples.

|    | IMDB | SST-2 | SST-5 | TREC | YELP-2 | YELP-5 |
|----|------|-------|-------|------|--------|--------|
| 5  | 78.4±2.2 | 81.2±1.1 | 40.6±1.7 | 82.9±1.4 | 82.4±1.3 | 46.7±2.2 |
| 10 | 79.6±2.1 | 82.3±1.9 | 42.5±1.5 | 84.8±1.8 | 83.2±1.5 | 48.4±2.7 |
| 15 | 80.2±3.6 | 82.7±2.4 | 42.9±1.9 | 85.2±2.6 | 83.6±2.1 | 48.8±3.2 |
| 20 | 80.4±3.8 | 82.8±2.6 | 43.0±2.2 | 85.4±2.8 | 83.8±2.3 | 48.9±3.3 |

**On the augmentation and selection strategy.** The following experiments aim to evaluate the effectiveness of the proposed VMM and CSM. For comparison purposes, we substitute VMM with the traditional masking-then-filling (MLM) approach, and we explore three strategies for CSM: Random selection, Mahalanobis and Cosine-based similarity selection. Results in Table 9 reveal that VMM outperforms MLM, with the latter potentially introducing conflicting tokens. Additionally, CSM outperforms other selection strategies, with the Mahalanobis-distance based approach as the runner-up. However, the random selection strategy introduces variability in the difficulty of chosen samples, leading to a larger standard deviation.

Table 9: Ablation study on different strategies for augmenting samples or selecting final samples.

| Dataset | MLM | Random | Mahalanobis | Cosine | M4DA |
|---------|-----|--------|-------------|--------|------|
| IMDB   | 79.0±2.8 | 77.9±4.6 | 78.8±2.6 | 78.3±3.4 | 79.6±2.1 |
| SST-2  | 81.7±3.1 | 80.5±3.4 | 81.4±2.0 | 80.9±1.6 | 82.3±1.9 |
| SST-5  | 41.9±2.3 | 39.6±2.9 | 41.1±1.8 | 40.4±0.9 | 42.5±1.5 |
| TREC   | 84.0±2.3 | 83.0±4.8 | 84.0±2.6 | 83.6±2.5 | 84.8±1.8 |
| YELP-2 | 82.6±2.5 | 81.2±2.5 | 82.4±0.8 | 81.8±2.2 | 83.2±1.5 |
| YELP-5 | 47.7±3.6 | 45.8±3.6 | 47.0±2.9 | 46.4±2.3 | 48.4±2.7 |

**On the Computational Complexity.** In our method, the primary computational overhead arises from the CSM module, where the Kantorovich-Rubinstein distance is computed with a complexity of $\mathcal{O}(p^3 \log(p))$ [Altschuler et al., 2017]. Here, $p$ represents the number of unique tokens from the input text. Yet, the computational complexity of M4DA is comparable to existing methods such as DoubleMix and TAA, as shown in Table 10. For instance, for DoubleMix, each sample is augmented $N$ times, resulting in $N$ augmented samples used for model fine-tuning. Conversely, M4DA selects the most challenging augmentation per sample. TAA, on the other hand, employs the Sequential Model-based Global Optimization technique to optimize the augmentation policy. This process involves accumulating the

size of the observation history and incurs significant computational costs for evaluating policies and updating the surrogate model, making it very time-consuming.

Table 10: Computational efficiency of various methods for training each epoch's samples in terms of GPU calculation time (seconds).

| Method | IMDB | SST2 | SST5 | TREC | YELP2 | YELP5 | Average |
|--------|------|------|------|------|-------|-------|---------|
| TAA | 1430 | 491 | 375 | 298 | 957 | 688 | 707 |
| DoubleMix | 328 | 114 | 303 | 128 | 337 | 658 | 311 |
| **M4DA** | 216 | 86 | 206 | 85 | 189 | 447 | 204 |

# 7 CONCLUSION

This paper introduces a novel augmentation method, `Mask-for-Data Augmentation` (**M4DA**), which strategically masks tokens from original samples for augmentation. Specifically, **M4DA** incorporates the Variance-Oriented Masker Module to introduce randomness by masking existing tokens in input sequences, thereby generating augmented samples with increasing variances. Additionally, we introduce the Complexity-Enhanced Selection Module with the goal of selecting the augmented sample with the highest semantic complexity. Extensive experiments demonstrate the superiority of **M4DA** over baseline methods in six highly-competitive benchmarks, encompassing scenarios with both limited and imbalanced training resources. Our method consistently outperforms existing approaches, showcasing its robustness and efficacy across diverse settings. In future work, we aim to explore more effective Data Augmentation techniques and extend the application of our approach to other downstream tasks.

**Acknowledgements**

This work is partially supported by the Australian Research Council Discovery Project (DP210101426), the Australian Research Council Linkage Project (LP200201035), AEGiS Advance Grant(888/008/268, University of Wollongong), Telstra-UOW Hub for AIOT Solutions Seed Funding (2024), Chinese Ministry of Education Humanities and Social Sciences General project (23YJAZH082), and Hubei Province education science planning key project (2022GA046).

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

# Appendix

**Xun Yao**[1]     **Zijian Huang**[1]     **Xinrong Hu**[1]     **Jie (Jack) Yang**[2]     **Yi Guo**[3]

[1] School of Computer Science and Artificial Intelligence, Wuhan Textile University, Hubei Province, China
[2] School of Computing and Information Technology, University of Wollongong, New South Wales, Australia
[3] School of Computer, Data and Mathematical Sciences, Western Sydney University, New South Wales, Australia

## A    EFFECT OF AUGMENTATIONS

We also computed the metrics on the IMDB and YELP-2 datasets, and the results are presented in Fig. 5 and Fig. 6. Color is employed to distinguish between different augmentation methods. For instance, the VII method (GPT3Mix, as shown in Table 1) is represented by the Blue color, while the IX (TreeMix) method is denoted by Red. Correspondingly, the index column in Table 1 corresponds to the method ID utilized in Fig. 1. Specifically, for each original sample $X^{\mathrm{ORG}}$, using each method mentioned in Table 1 generates an augmented sample $X^{\mathrm{AUG}}$. The augmented samples and original samples are simultaneously used to train the naive model $\mathcal{M}$. The augmented model $\mathcal{M}'$ is then tested on the training set.

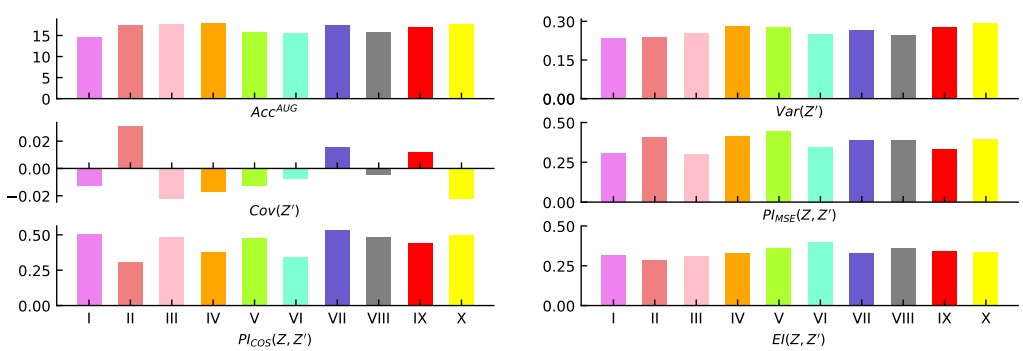

Figure 5: The resulting metrics on the IMDB dataset. The **naive** model $\mathcal{M}$ achieves the classification accuracy of 58.3±6.2, $\mathrm{Var}(Z)$ is 0.078±0.102, and $\mathrm{Cov}(Z)$ is 0.371±0.071. $Acc^{\mathrm{AUG}}$, $\mathrm{Var}(Z')$, and $\mathrm{Cov}(Z')$ respectively represent the differences between the augmented model **augmented** model $\mathcal{M}'$ and the **naive** model $\mathcal{M}$.

## B    F1 SCORE IN IMBALANCED SCENARIOS

To comprehensively evaluate the imbalance scenarios, we used the F1 score metric, with the results shown in Table 11 and Table 12. The following observations were noted: (1). All methods seemingly perform better in less imbalanced scenarios, similar to the Accuracy (%) metric. For example, in SST-2, the average F1 score increases by 5.7% and 7.4%, respectively, when the imbalance ratio rises from 2% to 5%; and (2). Our method significantly boosts the F1 score by an average of approximately 11.6% and 14.6%, respectively, outperforming other methods. These findings highlight the effectiveness of our method in handling imbalanced scenarios.

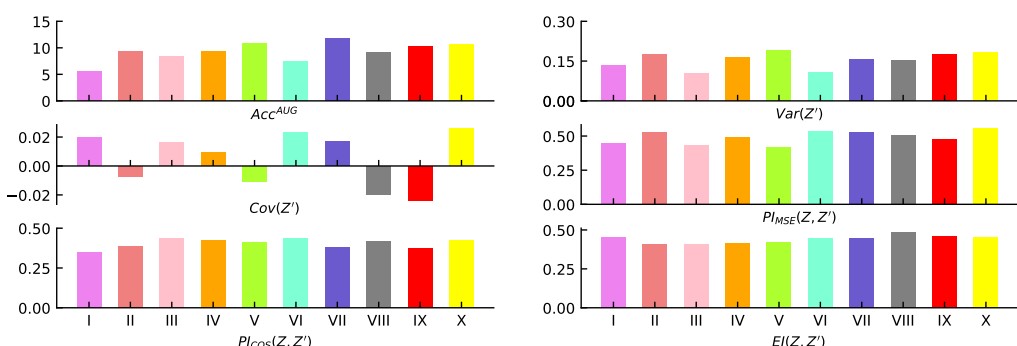

Figure 6: The resulting metrics on the YELP-2 dataset. The **naive** model $\mathcal{M}$ achieves the classification accuracy of 69.6±1.5, $\mathrm{Var}(Z)$ is 0.186±0.094, and $\mathrm{Cov}(Z)$ is 0.359±0.015. $Acc^{\mathrm{AUG}}$, $\mathrm{Var}(Z')$, and $\mathrm{Cov}(Z')$ respectively represent the differences between the augmented model **augmented** model $\mathcal{M}'$ and the **naive** model $\mathcal{M}$.

Table 11: Comparison of the classification F1 score performance under the **class-imbalance** setting. The best and second-best results are highlighted in bold and underline, respectively. Statistically significant gains achieved by the proposed method at $p$-values $< 0.01$ are marked with †.

| Method | SST-2 | | IMDB | | YELP-2 | |
|---|---|---|---|---|---|---|
| | $\gamma_{\mathrm{imb}} = 2\%$ | $\gamma_{\mathrm{imb}} = 5\%$ | $\gamma_{\mathrm{imb}} = 2\%$ | $\gamma_{\mathrm{imb}} = 5\%$ | $\gamma_{\mathrm{imb}} = 2\%$ | $\gamma_{\mathrm{imb}} = 5\%$ |
| BERT-Base | 61.3±0.8 | 62.7±2.3 | 63.8±0.6 | 67.2±2.1 | 67.5±0.7 | 69.5±1.4 |
| +OS | 64.9±1.2 | 68.1±1.8 | 67.4±1.5 | 71.7±3.6 | 68.3±1.0 | 71.9±1.7 |
| +EDA | 66.7±2.3 | 70.7±1.7 | 69.5±1.1 | 72.1±2.8 | 70.1±1.3 | 76.0±1.3 |
| +Back-trans | 67.8±2.9 | 71.4±1.6 | 67.1±1.2 | 69.7±1.7 | 72.6±2.0 | 79.1±2.6 |
| +BERT-aug | 68.4±2.4 | 73.3±2.2 | 72.3±1.1 | 74.1±1.9 | 72.7±1.1 | 82.1±2.1 |
| +AEDA | 67.2±1.1 | 75.7±2.7 | 69.9±0.7 | 75.5±2.8 | 71.2±2.5 | 83.6±1.4 |
| +TAA | 69.1±0.9 | 74.1±2.4 | 69.3±1.3 | 72.8±1.4 | 74.3±1.8 | 80.5±3.2 |
| +MTV | 65.9±1.7 | 72.6±2.2 | 68.7±1.7 | 72.0±1.1 | 72.0±1.2 | 77.4±2.1 |
| +Double Mix | 68.7±0.6 | 76.6±2.0 | 70.7±1.1 | 75.3±1.9 | 73.0±1.3 | 84.9±1.2 |
| +AWD | 68.7±1.2 | 77.0±1.4 | 71.0±1.7 | 76.0±1.7 | 74.1±2.4 | 86.0±1.4 |
| +SEMix | 68.9±1.5 | 76.9±2.6 | 71.3±0.9 | 75.8±1.9 | 73.9±1.9 | 85.3±2.5 |
| **+M4DA** | **70.8**±0.7† | **77.5**±1.8† | **73.0**±0.8† | **78.6**±1.6† | **76.1**±0.8† | **87.2**±1.8† |

Table 12: Comparison of the classification F1 score performance under the **class-imbalance** setting. The best and second-best results are highlighted in bold and underline, respectively. Statistically significant gains achieved by the proposed method at $p$-values $< 0.01$ are marked with †.

| Method | SST-2 | | IMDB | | YELP-2 | |
|---|---|---|---|---|---|---|
| | $\gamma_{\mathrm{imb}} = 2\%$ | $\gamma_{\mathrm{imb}} = 5\%$ | $\gamma_{\mathrm{imb}} = 2\%$ | $\gamma_{\mathrm{imb}} = 5\%$ | $\gamma_{\mathrm{imb}} = 2\%$ | $\gamma_{\mathrm{imb}} = 5\%$ |
| Roberta-Base | 63.7±0.6 | 69.8±1.2 | 62.6±1.8 | 64.2±2.1 | 64.2±1.4 | 70.2±1.2 |
| +OS | 65.2±2.5 | 73.2±2.6 | 64.0±2.0 | 72.6±1.8 | 69.7±1.2 | 74.3±2.2 |
| +EDA | 67.7±1.9 | 74.5±1.7 | 65.1±0.4 | 73.3±1.4 | 74.9±2.1 | 80.6±1.6 |
| +Back-trans | 69.0±2.7 | 75.1±1.8 | 70.3±0.8 | 75.1±0.9 | 78.1±1.4 | 82.0±0.9 |
| +BERT-aug | 70.3±1.5 | 76.7±2.0 | 69.5±2.6 | 74.5±1.7 | 72.9±1.8 | 81.3±1.8 |
| +AEDA | 73.0±0.4 | 80.8±2.2 | 71.8±1.5 | 78.4±0.6 | 75.1±1.9 | 82.4±2.2 |
| +TAA | 72.1±1.6 | 78.0±1.8 | 72.8±1.2 | 76.3±1.7 | 76.0±1.6 | 83.2±1.1 |
| +MTV | 71.2±1.8 | 76.6±0.7 | 70.6±2.8 | 76.9±0.8 | 75.2±1.7 | 80.3±1.3 |
| +Double Mix | 71.4±1.2 | 79.6±1.3 | 72.0±2.1 | 78.4±1.1 | 77.2±1.8 | 83.0±1.0 |
| +AWD | 72.7±1.4 | 81.8±2.2 | 72.5±1.2 | 80.9±1.0 | 79.3±2.2 | 84.6±0.8 |
| +SEMix | 72.0±1.7 | 81.2±1.4 | 72.2±1.3 | 80.4±1.3 | 79.5±2.1 | 84.1±1.2 |
| **+M4DA** | **74.1**±1.8† | **83.9**±1.2† | **73.8**±1.6† | **82.6**±1.5† | **81.2**±1.6† | **86.8**±1.3† |

## C  ON THE VARYING MODEL SIZES

We conducted an ablation study to evaluate the impact of varying model sizes, using RoBERTa-Large as the encoder while keeping all other configurations constant. The evaluation includes the methods BERT-aug, AEDA, Double-Mix, AWD, and SEMix, chosen for their demonstrated superior performance when using BERT-Base. The classification accuracy for low-resource and class-imbalance scenarios is presented in Table 13 and Table 14, respectively. Across these settings, our

method consistently delivered the highest performance across all datasets, validating its robust generalization capacity across different baselines and model sizes.

Table 13: Comparison of classification accuracy with the **low-resource** setting with the **RoBERTa-Large**. The best and the second best results are indicated in bold and underline, respectively. The first and second number represents the performance obtained from the low-resource and full dataset, respectively. Statistically significant gains achieved by the proposed method at $p$-values $< 0.01$ are marked with †.

| Method | IMDB | SST-2 | SST-5 | TREC | YELP-2 | YELP-5 |
|---|---|---|---|---|---|---|
| RoBERTa-Large | 77.3±1.8 | 76.8±2.2 | 37.2±2.5 | 78.0±3.6 | 84.3±2.9 | 44.8±3.8 |
| +BERT-aug | 86.8±2.7 | 84.5±3.1 | 40.6±4.0 | 83.3±3.9 | 91.6±2.6 | 47.2±5.2 |
| +AEDA | 84.3±2.0 | 85.4±2.9 | 42.3±3.5 | 85.6±5.5 | 89.1±2.4 | 48.9±4.0 |
| +Double Mix | 86.4±2.3 | 86.2±2.4 | 44.9±2.1 | 84.8±2.2 | 91.0±1.9 | 51.3±2.2 |
| +AWD | 88.0±3.8 | 88.6±3.1 | 46.7±3.8 | 85.1±2.4 | 93.0±2.8 | 50.8±3.2 |
| +SEMix | 87.6±2.7 | 86.9±2.6 | 47.2±1.1 | 84.2±3.7 | 92.2±2.9 | 49.6±3.5 |
| **+M4DA** | **88.9**±1.6† | **89.5**±2.3† | **48.4**±1.6† | **86.7**±2.0† | **94.1**±1.8† | **52.3**±2.1† |

Table 14: Comparison of the classification performance with the **class-imbalance** setting with the **RoBERTa-Large**. The best and second-best results are highlighted in bold and underline, respectively. Statistically significant gains achieved by the proposed method at $p$-values $< 0.01$ are marked with †.

| Method | SST-2 | | IMDB | | YELP-2 | |
|---|---|---|---|---|---|---|
| | $\gamma_{\text{imb}} = 2\%$ | $\gamma_{\text{imb}} = 5\%$ | $\gamma_{\text{imb}} = 2\%$ | $\gamma_{\text{imb}} = 5\%$ | $\gamma_{\text{imb}} = 2\%$ | $\gamma_{\text{imb}} = 5\%$ |
| RoBERTa-Large | 53.4±1.3 | 59.6±2.7 | 51.8±1.1 | 57.2±1.9 | 54.8±1.6 | 59.2±1.9 |
| +OS | 55.1±1.9 | 62.8±3.0 | 52.2±1.4 | 59.9±2.2 | 55.6±1.3 | 61.5±1.6 |
| +BERT-aug | 64.4±3.3 | 75.5±4.3 | 61.7±3.1 | 71.0±3.4 | 67.3±3.4 | 80.6±5.4 |
| +AEDA | 71.1±5.1 | 77.3±5.4 | 63.2±2.7 | 75.5±3.8 | 70.8±2.9 | 84.7±3.9 |
| +Double Mix | 68.2±3.4 | 82.8±2.8 | 64.6±1.9 | 77.4±2.5 | 74.8±1.6 | 82.6±2.7 |
| +AWD | 72.3±2.1 | 82.6±3.1 | 68.1±2.0 | 79.8±2.8 | 78.4±1.7 | 87.6±3.2 |
| +SEMix | 73.0±3.7 | 83.5±2.8 | 66.9±2.6 | 80.3±3.6 | 75.1±2.4 | 84.0±4.3 |
| **+M4DA** | **74.2**±2.9† | **85.0**±2.7† | **69.3**±1.4† | **82.9**±2.8† | **80.3**±2.4† | **89.7**±3.8† |

# D   DISCUSSION ON THE MASKING RATE

The initial masking rate is set at 5%, and a condition with a minimum of one masked token per augmentation is also enforced. That is, we ensure the inclusion of at least one `[Mask]` in each augmentation. The original sample is omitted from augmentation in the extreme case where only a single token is present. Furthermore, the masking probability $\eta$ is entirely data driven, *i.e.*, dynamically calculated during model fine-tuning. Put simply, the value of $\eta$ is derived from $\omega$ (as illustrated in Eq. 4), while $\omega$ is optimized using Eq. 6. The loss function shows a balance between augmenting data variance and ensuring the model's predictions align closely with the ground-truth labels. Arguably, if the level of missingness is high, the loss incurred from model predictions will escalate, thereby preventing excessive $\eta$.