# OpenReview forum: "Masking the Unknown: Leveraging Masked Samples for Enhanced Data Augmentation"
_auai.org/UAI/2024/Conference — UAI 2024 poster_

### Official Review · Reviewer_s2W6 · 2024-03-02

**Q2-1 Originality-Novelty:** 3
**Q2-2 Correctness-Technical Quality:** 3
**Q2-5 Clarity Of Writing:** 3

**Q10 Ethical Concerns:**

No ethical concerns.

**Q1 Summary And Contributions:**

This paper studies data augmentation for NLP tasks. Based on the finding that increased variance brought by data augmentation could lead to performance improvement, the authors propose to conduct masking to only augment the tokens that could lead to maximum increase of variance. Moreover, a complexity-enhanced selection module is introduced to further improve the effectiveness of the augmentation. Through extensive experiments, the authors carefully validated the proposed method which surpasses other baseline methods by a considerable margin.

**Q2-3 Extent To Which Claims Are Supported By Evidence:**

2: Fair: the main claims are somewhat supported by evidence (but the experimental evaluation may be weak, or does not match entirely with the claims, important baselines may be missing, proofs contain important ideas but lack rigor, algorithmic details are only discussed superficially, references are imprecise, assumptions are not sufficiently motivated or explicated, etc.).

**Q2-4 Reproducibility:**

3: Good: key resources (e.g. proofs, code, data) are available and key details (e.g. proofs, experimental setup) are sufficiently well-described for competent researchers to confidently reproduce the main results.

**Q3 Main Strengths:**

-	This paper is well-written and easy to follow.
-	The general motivation is clear and convincing. The proposed methodology is straightforward and easy to implement.
-	The experimental results are good.

**Q4 Main Weakness:**

-	The masking module is automatically learned through optimization. However, it requires a predefined hyperparameter $\eta$ which decides how many tokens are chosen for augmentation. How the value of $\eta$ affects the performance is not discussed.
-	The motivation for conducting complexity-enhanced selection is not clear. The authors directly claim that random masking leads to arbitrary outcomes, but there is no empirical evidence or theoretical justification to support this claim.
-	There are learning parameters and masks to optimize in this method. How is computation efficiency compared to other baseline methods?

**Q5 Detailed Comments To The Authors:**

Please see weaknesses for details.

**Q9 Complying With Reviewing Instructions:**

Yes

---

> ### Author Rebuttal · Authors · 2024-04-09
>
> **Q1: The masking module is automatically learned through optimization[...] How the value of affects the performance is not discussed.**
>
> **A1:** The masking probability $\eta$ is completely $\textit{data-driven}$, $\textit{i.e.}$, meaning it's dynamically calculated during model fine-tuning. Specifically, the value of $\eta$ is derived from $\omega$ (as shown in Eq. (4)), while $\omega$ is optimized using Eq. (6). Moreover, the loss function ensures that $\textit{the variance of the embeddings of the selected augmentation in the feature space is maximized}$, aiming to enhance learning performance.
>
> **Q2: The motivation for conducting complexity-enhanced selection is not clear. The authors directly claim that random masking leads to arbitrary outcomes, but there is no empirical evidence or theoretical justification to support this claim.**
>
> **A2:** In the proposed M4DA, the VMM module augments original data by replacing existing tokens with $\texttt{[Mask]}$, while the CSM module selects the most challenging samples that minimize semantic coverage. Accordingly, the term "arbitrary outcomes" denotes uncontrolled sampling from VMM, without CSM selection. Furthermore, Figure 3 illustrates an empirical comparison between the model with VMM only and the full model. This comparison shows the significant contribution of VMM in improving model performance by selecting challenging samples, compared to VMM.
>
> **Q3: There are learning parameters and masks to optimize in this method. How is computation efficiency compared to other baseline methods?**
>
> **A3:** The primary computational overhead arises from the CSM module, where the Kantorovich-Rubinstein distance is computed with a complexity of $\mathcal{O}(p^3\log(p))$[1], where $p$ represents $\textit{the number of unique tokens}$ from the input text. This detail will be included in the revision.
>
> The computational complexity of M4DA is comparable to existing methods like DoubleMix and TAA. In DoubleMix, each sample is augmented $N$ times, resulting in $N$ augmented samples used for model fine-tuning. In contrast, M4DA selects the most challenging augmentation per sample. As for TAA, it utilizes the Sequential Model-based Global Optimization technique to optimize the augmentation policy. This involves accumulating the size of the observation history and incurring computational costs for evaluating policies and updating the surrogate model, which is very time-consuming. An empirical comparison is provided below, showing the computational time per epoch:
>
> | Second | IMDB | SST2 | SST5 | TREC | YELP2 | YELP5 | Avg |
> | ------------ | ---- | ---- | ---- | ---- | ----- | ----- | ---- |
> | TAA          | 1430  | 491  | 375  | 298  | 957   | 688   | 707  |
> | DoubleMix    | 328  | 114  | 303  | 128  | 337   | 658   | 311  |
> | **M4DA**     | **216**  | **86**   | **206**  | **85**   | **189**   | **447**   | **204**  |
>
> [1] J. Altschuler, et al, Near-linear time approximation algorithms for optimal transport via sinkhorn iteration, NeurIPS, 2017.

---

### Official Review · Reviewer_qDdd · 2024-03-17

**Q2-1 Originality-Novelty:** 2
**Q2-2 Correctness-Technical Quality:** 3
**Q2-5 Clarity Of Writing:** 3

**Q1 Summary And Contributions:**

This work first evaluates existing data augmentation methods on multiple text-classification benchmarks. Then, it proposes a new augmentation method named Mask-for-Data Augmentation (M4DA). It consists of Variance-Oriented Masker Module (VMM) and Complexity- Enhanced Selection Module (CSM). The effectiveness of the proposed M4DA is empirically evaluated on multiple text-classification datasets including IMDB, SST-2, SST-5, TREC and YELP-2/5.

**Q2-3 Extent To Which Claims Are Supported By Evidence:**

3: Good: the main claims are supported by convincing evidence (in the form of adequate experimental evaluation, proofs, (pseudo-)code, references, assumptions).

**Q2-4 Reproducibility:**

2: Fair: key resources (e.g. proofs, code, data) are unavailable but key details (e.g. proof sketches, experimental setup) are sufficiently well-described for an expert to confidently reproduce the main results.

**Q3 Main Strengths:**

1. This work performs an empirical study of existing data augmentation methods for text classification.

2. It introduces the [Mask]-for-Data-Augmentation approach, named as M4DA.

3. The M4DA shows strong performance on multiple text-classification datasets.

**Q4 Main Weakness:**

1. This work does not evaluate and compare with more recent augmentation methods.
- As shown in Table 1, this work compares with the methods up to 2022.
- More recent methods should be compared as follows, to name a few.
- S. Kwon et al., Explainability-Based Mix-Up Approach for Text Data Augmentation, TKDD 2023.
- G. Sahu et al., PromptMix: A Class Boundary Augmentation Method for Large Language Model Distillation, EMNLP 2023.
- B. Li et al., Mixpro: Simple yet effective data augmentation for prompt-based learning. 2023.
- J. Chen et al., Adversarial word dilution as text data augmentation in low-resource regime, AAAI 2023.
- H. Zheng et al., Self-Evolution Learning for Mixup: Enhance Data Augmentation on Few-Shot Text Classification Tasks, EMNLP 2023.

2. Recently, large language models are actively used for data augmentation purpose. Therefore, they should be discussed and compared empirically.
- For example, H. Dai et al., AugGPT: Leveraging ChatGPT for Text Data Augmentation, arXiv 2023.
- C. Whitehouse et al., LLM-powered Data Augmentation for Enhanced Crosslingual Performance, EMNLP 2023.

3. The proposed method is rather simple.
- The VMM is a simple random mask-based augmentation.
- The CSM is also a simple sample selection with the high semantic complexity. Its unique feature is to utilize the Kantorovich-Rubinstein Distance. However, its definition and optimization are borrowed from existing literature of [Villani, 2016] and [Cuturi, 2013].

**Q5 Detailed Comments To The Authors:**

Please refer to Q4 Main Weakness.
My final score will be adjusted according to the authors’ responses to my concerns.

**Q9 Complying With Reviewing Instructions:**

Yes

---

> ### Author Rebuttal · Authors · 2024-04-09
>
> **Q1: This work does not evaluate and compare with more recent augmentation methods[...] More recent methods should be compared as follows, to name a few.**
>
> **A1:** The revised version will integrate two recent baseline studies [1][2], as they also employ the same encoder (BERT-base) and utilized datasets. Their results are directly sourced from the original papers. Below, we present the comparison with M4DA, where $k$ denotes the number of training samples selected from each class:
>
> | Dataset ($k$)		| Method	| Result	|
> | ------------- | --------- | --------- |
> | TREC (20)			| [1]	| 83.7±4.4	|
> | 					| **M4DA**		| **84.8±1.8**	|
> | SST2 (10)			| [1]	| 65.4±6.8 |
> | 					| [2]		| 57.6     |
> | 					| **M4DA**		| **67.6±4.7**	|
> | SST2 (50)			| [1]	| 82.7±5.2	|
> | 					| **M4DA**		| **84.3±2.3**	|
>
> The experimental results demonstrate  the efficacy of M4DA. To provide a more comprehensive comparison, we will reproduce these two baselines across all datasets in the revision.
>
> [1] J. Chen et al., Adversarial word dilution as text data augmentation in low-resource regime, AAAI 2023.
>
> [2] H. Zheng et al., Self-Evolution Learning for Mixup: Enhance Data Augmentation on Few-Shot Text Classification Tasks, EMNLP 2023.
>
> **Q2: Recently, large language models are actively used for data augmentation purpose. Therefore, they should be discussed and compared empirically.**
>
> **A2:** Thanks for highlighting this direction. A literature review, including the two listed references [3][4], has been conducted. However, it's important to note that the proposed method is not directly comparable to [3][4]. For instance, while AugGPT[3] necessitates fine-tuning the model twice (first using a large base set of labeled samples, and then using augmented samples from the target set), M4DA only involves the latter part. Accordingly, M4DA is compared with a GPT-3.5 (ChatGPT-based) method [5] (this result is directly cited from the original paper), and the comparison (Acc for SST2 and Macro-F1 for TREC) is shown as below (where $k$ is the number of training samples chosen from each class):
>
>   | Dataset ($k$)		| Method				| Result	|
>   | ------------- | --------- | --------- |
>   | SST2 (10/20)			| GPT3.5-Par[5]		| 62.5/69.0	|
>   | 						| GPT3.5-Desc[5]		| 78.6/82.6	|
>   | 						| **M4DA**					| 67.6/75.2	|
>   | TREC (10/20)		     | GPT3.5-Par[5]		| 31.3/44.8	|
>   | 						| GPT3.5-Desc[5]		| 31.4/42.9	|
>   | 						|**M4DA**					| 58.6/72.1	|
>
> The performance shows M4DA achieves comparable performance, and outperforms [5] with the TREC dataset. Notably, with a much smaller model size, our method offers a cost-effective solution. However, the ChatGPT-based method may encounter many challenges, such as failing to respond due to sensitive content within the datasets.
>
> [3] H. Dai et al., AugGPT: Leveraging ChatGPT for Text Data Augmentation, arXiv 2023.
>
> [4] C. Whitehouse et al., LLM-powered Data Augmentation for Enhanced Crosslingual Performance, EMNLP 2023.
>
> [5] Frédéric Piedboeuf, Philippe Langlais, Is ChatGPT the ultimate Data Augmentation Algorithm? EMNLP2023.
>
> **Q3: The proposed method is rather simple[...] However, its definition and optimization are borrowed from existing literature of [Villani, 2016] and [Cuturi, 2013].**
>
> **A3:** We are appreciative of the reviewer's recognition of the clarity of our writing. Beneath its simplicity, we wish to offer some additional remarks hereafter. The VMM module is a masking module with $\textit{data-driven masking probability}$, $\textit{i.e.}$, the optimal masking probability is determined $\textit{automatically}$ by the dataset. The selection module, CSM, selects the most challenging instances in the input space based on the disparity in semantic coverage compared to the original input. In essence, the purpose of CSM is to locate the augmentation that is $\textit{semantically the most distinct}$ from the given input. However, there isn't a direct connection between the semantic distribution and embeddings. This gap is filled by our loss function, where $\textit{the variance of the embeddings of the selected augmentation in the feature space is maximized}$ to increase learning performance [6]. The choices of each step have been validated by extensive numerical verification and ablation studies. The technical subtlety lies beneath the simplicity for the purpose of clarity in communicating ideas.
>
> [6] T. Hastie et al. The elements of statistical learning: Data mining, inference, and prediction, Springer, 2009.

---

### Official Review · Reviewer_eTaS · 2024-03-17

**Q2-1 Originality-Novelty:** 3
**Q2-2 Correctness-Technical Quality:** 3
**Q2-5 Clarity Of Writing:** 4

**Q10 Ethical Concerns:**

No.

**Q1 Summary And Contributions:**

This paper addresses an important issue of data augmentation in NLP text classification tasks due to limited resources or class imbalance. The author(s) propose leveraging data masking to re-use existing samples and train downstream classifiers, and demonstrate improved classification accuracy with several empirical datasets against extensive benchmarks.

**Q2-3 Extent To Which Claims Are Supported By Evidence:**

3: Good: the main claims are supported by convincing evidence (in the form of adequate experimental evaluation, proofs, (pseudo-)code, references, assumptions).

**Q2-4 Reproducibility:**

2: Fair: key resources (e.g. proofs, code, data) are unavailable but key details (e.g. proof sketches, experimental setup) are sufficiently well-described for an expert to confidently reproduce the main results.

**Q3 Main Strengths:**

The paper is very well written and easy to follow. Using masking as a form of perturbation for data augmentation is both interesting and does conserve resources (e.g., running LLMs to generate new data). The author(s) provide good experimentation using 6 empirical datasets and showing improvements in classification accuracy compared against many benchmarks. The author(s) address an interesting question of data augmentation in text classification tasks, under limited resource and/or class imbalance scenarios.

**Q4 Main Weakness:**

See comments below.

**Q5 Detailed Comments To The Authors:**

Does the naive encoder $\mathcal{M}$ follow the same architecture as augmented model $\mathcal{M’}$?

How is the masking probability $\eta$ chosen? In some ways this is analogous to introducing missingness via MCAR (Missing Completely At Random). What happens when missingness exceeds a certain threshold (e.g., 30%), and how does it impact downstream classifier performance? Is there an optimal range?

For the experiments section, how do you handle potential biases from text samples that are extremely short? For example, say if your masking probability was 10% but your sentence is only three words (e.g. “Movie was good”)? This ties back to my previous question.

Table 2: the author(s) could consider adding median and range of input text length depending on skewness of input text length distributions.

Table 4: Why was the class imbalance setting only performed for 3 out of 6 datasets from the previous low-resource setting? Also, under class imbalance, % Accuracy is not always a good indicator of classification performance. The author(s) should consider providing additional metrics such as F1-score, PR-AUC, Balanced Accuracy, etc. to demonstrate superior performance of M4DA, especially for minority classes.

How does M4DA compare to classic techniques for addressing class imbalance, such as oversampling or class weights?

Other: please fix in-text citations and formatting (some citations are not properly enclosed in brackets) for readability.

Please provide anonymized code for reproducibility, if possible. I would have expected to at least see minimal working examples or pseudocode.

**Q9 Complying With Reviewing Instructions:**

Yes

---

> ### Author Rebuttal · Authors · 2024-04-09
>
> **Q1: Does the naive encoder follow the same architecture as augmented model?**
>
> **A1:** That is correct. The model architecture remains consistent, with the only difference being that $\mathcal{M}'$ is trained using both original and augmented samples, whereas $\mathcal{M}$ is solely fine-tuned using original samples.
>
> **Q2: How is the masking probability chosen[...] Is there an optimal range?**
>
> **A2:** The masking probability $\eta$ is entirely data driven, $\textit{i.e.}$, dynamically calculated during model fine-tuning. Put simply, the value of $\eta$ is derived from $\omega$ (as illustrated in Eq. (4)), while $\omega$ is optimized using Eq. (6). The loss function shows a balance between augmenting data variance and ensuring the model's predictions align closely with the ground-truth labels. Arguably, if the level of missingness is high, the loss incurred from model predictions will escalate, thereby preventing excessive $\eta$. Additionally, we conducted an analysis of $\eta$ across all datasets and training epochs, revealing its variation within certain ranges. This result is shown below:
>
>   |         | IMDB | SST2 | SST5 | TREC | YELP2 | YELP5 |
>   | ------- | ---- | ---- | ---- | ---- | ----- | ----- |
>   | Min     | 20 % | 26 % | 28 % | 28 % | 38 %  | 20 %  |
>   | Max     | 39 % | 43 % | 42 % | 44 % | 68 %  | 59 %  |
>   | Avg     | 32 % | 31 % | 37 % | 39 % | 56 %  | 40 %  |
>
> **Q3: For the experiments section, how do you handle potential biases from text samples that are extremely short[...] This ties back to my previous question.**
>
> **A3:** The initial masking rate is set at 5\%, and a condition with a minimum of one masked token per augmentation is also enforced. That is, we ensure the inclusion of at least one $\texttt{[Mask]}$ in each augmentation. The original sample is omitted from augmentation in the extreme case where only a single token is present. This implementation detail will be included into the revision.
>
> **Q4: Table 2: the author(s) could consider adding median and range of input text length depending on skewness of input text length distributions.**
>
> **A4:** The statistic is shown below and will be provided in the revision:
>
>   | Dataset | Min | Max | Median | Avg |
>   | ------ | ---------- | ---------- | ------ |------ |
>   | IMDB   | 2          | 2470       | 173    |326    |
>   | SST2   | 2          | 56         | 18     |18    |
>   | YELP2  | 1          | 1052       | 97     |139    |
>   | TREC   | 3          | 37         | 9      |10    |
>   | SST5   | 2          | 56         | 18     |19    |
>   | YELP5  | 1          | 1052       | 99     |141    |
>
> **Q5: Table 4: Why was the class imbalance setting only performed for 3 out of 6 datasets from the previous low-resource setting[...] How does M4DA compare to classic techniques for addressing class imbalance, such as oversampling or class weights?**
>
> **A5:** We limit our selection to three binary classification datasets, for the class imbalance setting, to adhere to the experiments outlined in [1], where samples from the negative class are considered the majority. The F1-score for all three imbalanced datasets, along with the results from oversampling (OS), are presented below. These updated results will be incorporated into the revision.
>
>  | Method  | SST2    | SST2    | IMDB      | IMDB    | YELP2   | YELP2     |
>   | ----------- | -- | - | -- | - | - | - |
>   | Acc / F1    |  $\gamma_{\mathrm{imb}}$=2%       | $\gamma_{\mathrm{imb}}$=5%        |  $\gamma_{\mathrm{imb}}$=2%       | $\gamma_{\mathrm{imb}}$=5%        |  $\gamma_{\mathrm{imb}}$=2%      | $\gamma_{\mathrm{imb}}$=5%        |
>   | BERT-base   | 50.2±1.3/61.3±0.8 | 55.0±6.3/62.7±2.3 | 50.1±0.0/63.8±0.6 | 51.8±4.6/67.2±2.1 | 52.2±1.2/67.5±0.7 | 56.2±2.6/69.5±1.4 |
>   | +OS         | 52.0±1.8/64.9±1.2 | 58.6±5.6/68.1±1.8 | 52.3±2.7/67.4±1.5 | 59.1±6.3/71.7±3.6 | 54.4±2.6/68.3±1.0 | 61.3±3.1/71.9±1.7 |
>   | +BERT-aug  | 54.6±3.2/68.4±2.4 | 64.9±3.7/73.3±2.2 | 61.2±2.2/72.3±1.1 | 68.5±3.8/74.1±1.9 | 63.7±1.6/72.7±1.1 | 79.4±3.3/82.1±2.1 |
>   | +AEDA      | 53.4±1.8/67.2±1.1 | 67.8±3.9/75.7±2.7 | 57.8±1.2/69.9±0.7 | 70.2±5.4/76.6±2.8 | 60.6±1.7/71.2±2.5 | 80.2±4.0/83.6±1.4 |
>   | +TAA        | 56.5±3.7/69.1±0.9 | 66.1±4.9/74.1±2.4 | 56.9±2.8/69.5±1.3 | 66.7±4.2/72.8±1.4 | 65.8±2.8/74.3±1.8 | 77.5±4.5/80.5±3.2 |
>   | +DoubleMix | 55.1±2.5/68.7±0.6 | 68.5±3.9/76.6±2.0 | 59.3±2.1/70.7±1.1 | 69.6±4.0/75.3±1.9 | 63.8±1.9/73.0±1.3 | 81.9±2.7/84.9±1.2 |
>   | +**M4DA**|**58.4±1.6/70.8±0.7**|**70.3±4.4/77.5±1.8**|**61.2±2.5/73.0±0.8**|**73.4±3.8/78.6±1.6**|**67.4±1.3/76.1±0.8**| **84.9±3.1/87.2±1.8**|
>
> [1] S. Ren et al, Text AutoAugment: Learning Compositional Augmentation Policy for Text Classification, EMNLP 2021.
>
> **Q6: Please provide anonymized code for reproducibility, if possible. I would have expected to at least see minimal working examples or pseudocode.**
>
> **A6:** The code is available at https://anonymous.4open.science/r/M4DA-25B2.

---

### Official Review · Reviewer_bioK · 2024-03-21

**Q2-1 Originality-Novelty:** 2
**Q2-2 Correctness-Technical Quality:** 2
**Q2-5 Clarity Of Writing:** 3

**Q1 Summary And Contributions:**

The authors research text data augmentation in this paper and propose a variance-based augmentation approach. Experiments are conducted on several text classification dataset with many baselines.

**Q2-3 Extent To Which Claims Are Supported By Evidence:**

2: Fair: the main claims are somewhat supported by evidence (but the experimental evaluation may be weak, or does not match entirely with the claims, important baselines may be missing, proofs contain important ideas but lack rigor, algorithmic details are only discussed superficially, references are imprecise, assumptions are not sufficiently motivated or explicated, etc.).

**Q2-4 Reproducibility:**

3: Good: key resources (e.g. proofs, code, data) are available and key details (e.g. proofs, experimental setup) are sufficiently well-described for competent researchers to confidently reproduce the main results.

**Q3 Main Strengths:**

1. Text data augmentation is one of the fundmental but practical question to research for NLP.
2. Multiple baselines are involved which is good to see.

**Q4 Main Weakness:**

1. I don't quite get the motivation for using the Kantorovich-Rubinstein Distance. What is the relation for using this distance measurment to the variouse-oriented proposal as discussed before?
2. The authors mention that previous augmentation approaches may result in semantically meaningless or conflicting augmentations. How does the variance-based approach specificly deal with these weaknesses? The motivation and the proposed method seem to be a little bit inconsistent to me.
3. In experiments the authors only include BERT and RoBERT and they are all encoder-only architecture (bi-directional). Experiments can be further riched by including some decoder-only and encoder-decoder models.
4. Right now there are many tasks need to have long text length in their input. How does this approach scale to long text length, e.g., 2,048 or 4.096? Does the time complexity of this approach related to the text lenght, and if so, what is it?

**Q5 Detailed Comments To The Authors:**

1. In Figure 1, what does the colors mean? There is no caption for this figure.
2. Any meaning for the index column in Table 1?

**Q9 Complying With Reviewing Instructions:**

Yes

---

> ### Author Rebuttal · Authors · 2024-04-09
>
> **Q1: I don't quite get the motivation for using the Kantorovich-Rubinstein Distance. What is the relation for using this distance measurment to the variouse-oriented proposal as discussed before?**
>
> **A1:** The reason for using Kantorovich-Rubinstein (KR) distance is to calculate the semantic coverage, which is defined to encompass all possible joint distributions and hence is comprehensive. Accordingly, the CSM module utilizes the KR distance to select the hardest sample $\textit{within the semantic space}$, $\textit{i.e.}$, the sample with the minimal semantic overlap with the original sample. Furthermore, via optimizing the loss in Eq. (6), we ensure that their representations $\textit{within the feature space}$ from the encoder exhibit maximum variance. This is an designed alignment to facilitate a direct connection between semantic and feature spaces, effectively bridging the gap between them.
>
> **Q2: The authors mention that previous augmentation approaches may result in semantically meaningless or conflicting augmentations[...] The motivation and the proposed method seem to be a little bit inconsistent to me.**
>
> **A2:** Prior approaches often incorporate synonym replacement along with random insertion, swapping, or deletion of words, which could produce meaningless augmentation. The variance-based approach of M4DA aims to enhance the diversity of augmented data while still maintaining semantic relevance. By substituting original input tokens with $\texttt{[Mask]}$, a crucial aspect of encoder pretraining, we ensure that the semantic coherence of the augmented data is preserved. In other words, during pretraining, the encoder acquires the ability to predict masked tokens based on the surrounding context, thereby gaining the understanding of the semantic relationships within the data. Consequently, with the proposed M4DA, the model leverages this pre-established semantic knowledge embedded within the encoder, ensuring the semantic integrity and coherence of augmented data.
>
> **Q3: In experiments the authors only include BERT and RoBERT and they are all encoder-only architecture (bi-directional). Experiments can be further riched by including some decoder-only and encoder-decoder models.**
>
> **A3:** We acknowledge the potential benefits of incorporating a wider variety of models, including decoder-only and encoder-decoder architectures. This paper predominantly employs encoder-only architectures, while this choice stems from our primary aim: to assess the efficacy of M4DA within the context of widely adopted pre-trained language models, many of which are built upon encoder-only designs. In future research, we plan to explore encoder-decoder architectures, as outlined in [1], for machine translation tasks with M4DA.
>
> [1] Z. Liu et al, Improving Text Generation with Dynamic Masking and Recovering, IJCAI 2021.
>
> **Q4: Right now there are many tasks need to have long text length in their input[...] Does the time complexity of this approach related to the text lenght, and if so, what is it?**
>
> **A4:** The proposed M4DA comprises two modules: VMM, which masks existing input tokens, and CSM, which selects augmented samples with minimum semantic coverage. The primary computational overhead arises from the CSM module, where the Kantorovich-Rubinstein distance is computed with a complexity of $\mathcal{O}(p^3\log(p))$ [2], and $p$ represents *the number of unique tokens from the input text*. This detail will be included in the revision.
>
> [2] J. Altschuler, et al, Near-linear time approximation algorithms for optimal transport via sinkhorn iteration, NeurIPS, 2017.
>
> **Q5: In Figure 1, what does the colors mean[...] Any meaning for the index column in Table 1?**
>
> **A5:** Color is employed to distinguish between different augmentation methods. For instance, the VII method (GPT3Mix, as shown in Table 1) is represented by the Blue color, while the IX (TreeMix) method is denoted by Red. Correspondingly, the index column in Table 1 corresponds to the method ID utilized in Figure 1.

---

### Official Review · Reviewer_x7id · 2024-03-31

**Q2-1 Originality-Novelty:** 3
**Q2-2 Correctness-Technical Quality:** 3
**Q2-5 Clarity Of Writing:** 4

**Q1 Summary And Contributions:**

This work proposes a novel data augmentation approach motivated by experimental observations. The authors found that data variance plays a crucial role in data augmentation. Thus, the authors propose a novel mask approach. Some experiments are conducted to verify the effectivenss.

**Q2-3 Extent To Which Claims Are Supported By Evidence:**

3: Good: the main claims are supported by convincing evidence (in the form of adequate experimental evaluation, proofs, (pseudo-)code, references, assumptions).

**Q2-4 Reproducibility:**

3: Good: key resources (e.g. proofs, code, data) are available and key details (e.g. proofs, experimental setup) are sufficiently well-described for competent researchers to confidently reproduce the main results.

**Q3 Main Strengths:**

The paper is well written and proposes the innovative Mask-for-Data Augmentation (M4DA) scheme for enhancing data augmentation strategies in natural language processing tasks, enabling a more diverse and robust training dataset.

This paper proposes a new method and verifies the effectiveness of M4DA through experiments on multiple text classification benchmarks.

These experimental results provide strong evidence to support the performance advantages of the M4DA method in practical applications.

A detailed ablation study was carried out, analyzing the contribution of two key components, the VarianceOriented Masker Module (VMM) and the ComplexityEnhanced Selection Module (CSM), to overall performance. This analysis helps to understand the role of each component, guiding future improvements and optimizations.

**Q4 Main Weakness:**

Some problems must be solved before it is considered for publication. If the following problems are well-addressed, this reviewer believes that the essential contribution of this paper is important for NLP.

1. This paper mainly focuses on text classification tasks, and there is a lack of discussion on the applicability and effect of M4DA methods in other types of NLP tasks (such as machine translation, speech recognition, etc.). This may limit the generalization and scope of application of the method.

2. Papers may not adequately account for the characteristics and difficulty of different datasets or may not include sufficient baseline methods in the experimental setup for comparison. In addition, if the effects of different model sizes or training strategies are not taken into account in the experiment, this may also affect the interpretation and generalization of the experimental results.

3. Although experiments verify the effectiveness of M4DA, the theoretical explanation of why the masking strategy can effectively improve the model's performance may not be sufficient, and the specific mechanism of the increase of data variance on the model's performance is not fully discussed. More in-depth theoretical analysis and interpretability studies will help enhance the credibility and user acceptance of the methodology.

**Q5 Detailed Comments To The Authors:**

See Main Weakness.

**Q9 Complying With Reviewing Instructions:**

Yes

---

> ### Author Rebuttal · Authors · 2024-04-09
>
> **Q1: This paper mainly focuses on text classification tasks[...] This may limit the generalization and scope of application of the method.**
>
> **A1:** This paper primarily targets text classification tasks, which are widely acknowledged as typical downstream challenges in NLP. Other applications, such as sentiment analysis, extractive summarization, and intent detection inherently, manifest as text classification problems. Therefore, it is interesting to consider the applicability and potential extensions of M4DA to these tasks and possibly to other domains like machine translation. We acknowledge this as a significant area and intend to incorporate it into our future work.
>
> **Q2: Papers may not adequately account for the characteristics and difficulty[...] generalization of the experimental results.**
>
> **A2:** The revision will incorporate two recent baseline studies [1][2], with their results directly cited from the original papers. Below is the comparison with M4DA, delineated by $k$, indicating the number of training samples chosen from each class:
>
> |Dataset ($k$)|Method|Result|
> |-|-|-|
> |TREC (20) |[1]|83.7±4.4|
> ||**M4DA**|**84.8±1.8**|
> |SST2 (10)|[1]|65.4±6.8|
> ||[2] |57.6|
> ||**M4DA**|**67.6±4.7**|
> |SST2 (50)| [1]| 82.7±5.2|
> ||**M4DA**|**84.3±2.3**|
>
> Furthermore, an additional ablation study assessing the impact of varying model sizes is conducted, utilizing Roberta-Large as the encoder (keeping the rest of configuration consistent). For the time being, the evaluation includes the methods of BERT-aug, AEDA, and Double-Mix, chosen for their demonstrated superior performance when using BERT-base. Three datasets are taken into account, and their classification accuracy under the low-resource setting is provided below:
>
>   |Method|IMDB ($k$=40)|SST2 ($k$=40)|YELP2 ($k$=40)|
>   |-|-|-|-|
>   |Roberta-Large|77.3±1.8|76.8±2.2|84.3±2.9|
>   |+BERT-aug|86.8±2.7|84.5±3.1|91.6±2.6|
>   |+AEDA |84.3±2.0 |85.4±2.9 |89.1±2.4|
>   |+Double-Mix|86.4±2.3| 86.2±2.4|91.0±1.9|
>   |+**M4DA**|**88.9±1.6**|**89.5±2.3**|**94.1±1.8**|
>
> In addition, the accuracy under the class-imbalance setting is listed below, where $\gamma_{\mathrm{imb}}$ is the ratio between positive and negative samples.
>
>   |Method|IMDB|IMDB|SST2|SST2|YELP2|YELP2|
>   |-|-|-|-|-|-|-|
>   ||$\gamma_{\mathrm{imb}}$=2%|$\gamma_{\mathrm{imb}}$=5% |  $\gamma_{\mathrm{imb}}$=2%| $\gamma_{\mathrm{imb}}$=5%|$\gamma_{\mathrm{imb}}$=2% | $\gamma_{\mathrm{imb}}$=5%|
>   |Roberta-Large|53.4±1.3|59.6±2.7|51.8±1.1|57.2±1.9|54.8±1.6|59.2±1.9|
>   |+BERT-aug|64.4±3.3|75.5±4.3|61.7±3.1|71.0±3.4|67.3±3.4|80.6±5.4|
>   |+AEDA|71.1±5.1|77.3±5.4|63.2±2.7|75.5±3.8|70.8±2.9|84.7±3.9|
>   |+Double-Mix |68.2±3.4|82.8±2.8|64.6±1.9|77.4±2.5|74.8±1.6|82.6±2.7|
>   |+**M4DA**|**74.2±2.9**|**85.0±2.7**|**69.3±1.4**|**82.9±2.8**|**80.3±2.4**|**89.7±3.8**|
>
> The experimental results demonstrate the efficacy of M4DA, and the findings validate its robust generalization capacity across different baselines and model sizes. We will include all datasets and baselines in the revision.
>
> [1] J. Chen et al., Adversarial word dilution as text data augmentation in low-resource regime, AAAI 2023.
>
> [2] H. Zheng et al., Self-Evolution Learning for Mixup: Enhance Data Augmentation on Few-Shot Text Classification Tasks, EMNLP 2023.
>
> **Q3: Although experiments verify the effectiveness of M4DA[...] user acceptance of the methodology.**
>
> **A3:** The theoretical analysis of M4DA may follow the similar route as in [3]. The difference is that M4DA replaces tokens by $\texttt{[Mask]}$. The VMM module proposes a range of altered perturbations (augmetations) based on the original input sequence and hence their corresponding convex hulls in the input space. Among these perturbations, the CSM module identifies the most challenging one in terms of semantic coverage relative to the original input. However, it's essential to note that the semantic distribution might differ in another space, potentially leading to unclear geometries. Our loss function addresses this mismatch by maximizing the variance of the selected perturbation in the feature space, where representations reside, thereby exposing the encoder to maximal variation to enhance learning performance [4].
>
> There are many aspects worth exploring theoretically, for example, the sensitivity of the encoder's learning capacity to the altered perturbations for the final representation. This is relatively linked to the encoder's universal approximation capability. Given the current fixed structures of encoders, our analysis is confined to these structures and adopts a probing approach. However, based on M4DA, a comprehensive extension is warranted, as we have presented substantial empirical evidence supporting further investigation into these theoretical aspects.
>
> [3] X. Hu et al. [MASK] Insertion: a robust method for anti-adversarial attacks, EACL 2023.
>
> [4] T. Hastie et al. The elements of statistical learning: Data mining, inference, and prediction, Springer, 2009.

---

### Meta-Review · Area_Chair_U7kc · 2024-04-17

The paper studies the problem of data augmentation in masked language models. To improve the performance and efficiency, the paper develops a simple yet effective data augmentation method that selects masked tokens based on data variance. The proposed method has been tested on several text classification benchmarks, demonstrating significant improvements.

Reviewers appreciated the simplicity of the proposed method and its substantial improvements. During the discussion, the authors provided additional experimental results to further demonstrate the effectiveness of the proposed method. I encourage the authors to revise the manuscript according to the reviewers’ feedback.